# Biological methane production under putative Enceladus-like conditions

Ruth-Sophie Taubner[1,2], Patricia Pappenreiter[3], Jennifer Zwicker[4], Daniel Smrzka[4], Christian Pruckner[1], Philipp Kolar[1], Sébastien Bernacchi[5], Arne H. Seifert[5], Alexander Krajete[5], Wolfgang Bach[6], Jörn Peckmann [4,7], Christian Paulik [3], Maria G. Firneis[2], Christa Schleper[1] & Simon K.-M.R. Rittmann [1]

The detection of silica-rich dust particles, as an indication for ongoing hydrothermal activity, and the presence of water and organic molecules in the plume of Enceladus, have made Saturn's icy moon a hot spot in the search for potential extraterrestrial life. Methanogenic archaea are among the organisms that could potentially thrive under the predicted conditions on Enceladus, considering that both molecular hydrogen ($H_2$) and methane ($CH_4$) have been detected in the plume. Here we show that a methanogenic archaeon, *Methanothermococcus okinawensis*, can produce $CH_4$ under physicochemical conditions extrapolated for Enceladus. Up to 72% carbon dioxide to $CH_4$ conversion is reached at 50 bar in the presence of potential inhibitors. Furthermore, kinetic and thermodynamic computations of low-temperature serpentinization indicate that there may be sufficient $H_2$ gas production to serve as a substrate for $CH_4$ production on Enceladus. We conclude that some of the $CH_4$ detected in the plume of Enceladus might, in principle, be produced by methanogens.

[1] Archaea Biology and Ecogenomics Division, Department of Ecogenomics and Systems Biology, Universität Wien, 1090 Vienna, Austria. [2] Department of Astrophysics, Universität Wien, 1180 Vienna, Austria. [3] Institute for Chemical Technology of Organic Materials, Johannes Kepler Universität Linz, 4040 Linz, Austria. [4] Department of Geodynamics and Sedimentology, Center for Earth Sciences, Universität Wien, 1090 Vienna, Austria. [5] Krajete GmbH, 4020 Linz, Austria. [6] Geoscience Department, Universität Bremen, 28359 Bremen, Germany. [7] Institute for Geology, Center for Earth System Research and Sustainability, Universität Hamburg, 20146 Hamburg, Germany. Correspondence and requests for materials should be addressed to S.K.-M.R.R. (email: simon.rittmann@univie.ac.at)

Saturn's icy moon Enceladus emits jets of mainly water ($H_2O$) from its south-polar region[1]. Besides $H_2O$, the ion and neutral mass spectrometer (INMS) onboard NASA's Cassini probe detected methane ($CH_4$), carbon dioxide ($CO_2$), ammonia ($NH_3$), molecular nitrogen ($N_2$), and molecular hydrogen ($H_2$) in the plume[2]. In addition, carbon monoxide (CO) and ethene ($C_2H_4$) were found among other substances with moderate ambiguity[3–6] (Table 1). At $1608.3 \pm 4.5\,kg\,m^{[-3]}$, Enceladus possesses a relatively high-bulk density for an icy moon, which leads to the assumption that a substantial part of its core consists of chondritic rocks[7]. At the boundary between the liquid water layer and the rocky core, geochemical interactions are assumed to occur at low to moderate temperatures (<100 °C)[2,7,8]. The most prominent potential source of $H_2$ in Enceladus' interior may be oxidation of native and ferrous iron in the course of serpentinization of olivine in the chondritic core. Olivine hydrolysis at low temperatures is a key process for sustaining chemolithoautotrophic life on Earth[9] and if $H_2$ is produced in significant amounts on Enceladus, then it could also serve as a substrate for biological $CH_4$ production. Considering that $139 \pm 28 \times 10^9$ to $160 \pm 43 \times 10^9\,kg$ carbon year$^{-1}$ of the $CH_4$ found in the atmosphere of Earth is emitted from natural sources[10], including biological methanogenesis, the question was raised if $CH_4$ detected in the plume of Enceladus could in principle also originate from biological activity[11].

To date, methanogenic archaea are the only known microorganisms that are capable of performing biological $CH_4$ production in the absence of oxygen[12,13]. On Earth, methanogens are found in a wide range of pH (4.5–10.2), temperatures (<0–122 °C), and pressures (0.005–759 bar)[13] that overlap with conditions predicted in Enceladus' subsurface ocean, i.e., temperatures between 0 and above 90 °C[8], pressures of 40–100 bar[8], a pH between 8.5–10.5[8] and 10.8–13.5[14], and a salinity in the range of our oceans. While autotrophic, hydrogenotrophic methanogens might metabolise some of the compounds found in Enceladus' plume, other compounds which were detected in the plume with different levels of ambiguity, such as formaldehyde ($CH_2O$), methanol ($CH_3OH$), $NH_3$, CO, and $C_2H_4$ are known to inhibit growth of methanogens on Earth at certain concentrations[15–17].

Here we show that methanogens can produce $CH_4$ under Enceladus-like conditions, and that the estimated $H_2$ production rates on this icy moon can potentially be high enough to support autotrophic, hydrogenotrophic methanogenic life.

## Results

**Effect of gaseous inhibitors on methanogens**. To investigate growth of methanogens under Enceladus-like conditions, three thermophilic and methanogenic strains, *Methanothermococcus okinawensis* (65 °C)[18], *Methanothermobacter marburgensis* (65 °C)[19], and *Methanococcus villosus* (80 °C)[20], all able to fix carbon and gain energy through the reduction of $CO_2$ with $H_2$ to form $CH_4$, were investigated regarding growth and biological $CH_4$ production under different headspace gas compositions (Table 2) on $H_2/CO_2$, $H_2/CO$, $H_2$, Mix 1 ($H_2$, $CO_2$, CO, $CH_4$, and $N_2$) and Mix 2 ($H_2$, $CO_2$, CO, $CH_4$, $N_2$, and $C_2H_4$). These methanogens were prioritised due to their ability to grow (1) in a temperature range characteristic for the vicinity of hydrothermal vents[21], (2) in a chemically defined medium[22], and (3) at low partial pressures of $H_2$[23]. Also, in the case of *M. okinawensis*, the location of isolation was taken into consideration, since the organism was isolated from a deep-sea hydrothermal vent field at Iheya Ridge in the Okinawa Trough, Japan, at a depth of 972 m below sea level[18], suggesting a tolerance toward high pressure.

### Table 1 Compilation of Cassini's INMS data on Enceladus' plume composition over the last decade

| Species[a] | Volume mixing ratio | | | | | |
|---|---|---|---|---|---|---|
| | Waite et al. 2006[3] | Waite et al. 2009[4] | Waite et al. 2011[25] | Perry et al. 2015[26] | Bouquet et al. 2015[5] | Waite et al. 2017[b,2] |
| $H_2O$ | 90.7-91.5 | 90.0 ± 1.0 | **92.0 ± 3.0** | >90 | 87 | 96–99 |
| $CO_2$ | 3.14-3.26 | 5.3 ± 0.1 | 0.8 ± 0.3 | **0.6 ± 0.2** | 0.52 | 0.3-0.8 |
| CO | (3.29-4.27) | (4.4) | <1.5 | | ≤**0.64** | |
| $H_2$ | | (39) | <3.4 ± 1.0 | **1-5** | 11 | 0.4-1.4 |
| $CH_2O$ | | 0.31 ± 0.01 | **<0.032** | | | |
| $CH_3OH$ | | **0.015 ± 0.006** | 0.003 ± 0.002 | | | |
| $C_2H_4$ | | **<1.2** | | | | |
| $H_2S$ | | 0.0021 ± 0.0010 | 0.003 ± 0.001 | | **0.0021 ± 0.0010** | |
| $NH_3$ | | 0.82 ± 0.02 | 0.8 ± 0.03 | **0.9 ± 0.04** | 0.61 | 0.4-1.3 |
| $N_2$ | (3.29-4.27) | <1.1 | | | ≤**0.61** | |
| HCN | | <0.74 | 0.7 ± 0.3 | | ≤**0.12** | |
| $CH_4$ | 1.63-1.68 | 0.91 ± 0.05 | 0.21 ± 0.09 | **0.2 ± 0.1** | 0.19 | 0.1-0.3 |

[a] Values used in this study are marked in bold
[b] These recent observations based on the data of flyby E21 lead to the assumption that $H_2O$ is even more prominent, whereas the concentrations for the other major species ($NH_3$, $CO_2$, and $CH_4$) varied only slightly. The other components were categorised as minor species with moderate ambiguity (e.g., CO, $N_2$, $C_2H_4$, or $CH_2O$) or as potential species with high ambiguity (e.g., $H_2S$ or $CH_3OH$)[6]

### Table 2 Composition of the different test gases for the low-pressure experiments

| | $H_2$ (Vol.-%) | $CO_2$ (Vol.-%) | CO (Vol.-%) | $CH_4$ (Vol.-%) | $N_2$ (Vol.-%) | $C_2H_4$ (Vol.-%) |
|---|---|---|---|---|---|---|
| $H_2/CO_2$ | 80.097 | 19.903 | — | — | — | — |
| $H_2/CO$ | 80.290 | — | 19.710 | — | — | — |
| $H_2$ | 99.999 | — | — | — | — | — |
| Mix 1 | 22.900 | 19.490 | 27.790 | 14.430 | 15.390 | — |
| Mix 2 | 22.430 | 19.210 | 28.151 | 14.510 | 12.410 | 3.289 |

While *M. okinawensis*, *M. marburgensis*, and *M. villosus* all showed growth on $H_2/CO_2$ to similar optical densities, no growth of *M. marburgensis* could be observed when $C_2H_4$ (Mix 2) was supplied in the headspace (Fig. 1). Growth of both *M. villosus* and *M. okinawensis* was observed even when CO and $C_2H_4$ were both present in the headspace gas. However, while *M. villosus* showed prolonged lag phases and irregular growth under certain conditions, *M. okinawensis* grew stably and reproducibly on the different gas mixtures without extended lag phases (Fig. 1).

As expected, the final optical densities did not reach those of the experiments with $H_2/CO_2$, likely because in Mix 1 and Mix 2 lower absolute amounts of convertible gaseous substrate ($H_2/CO_2$) were available compared to the growth under pure $H_2/CO_2$. Consequently, growth kinetics showed a different, gas-limited linear inclination in the closed batch setup when using Mix 1 and Mix 2[22,24]. Due to its reproducible growth, *M. okinawensis* was chosen for more extensive studies on biological $CH_4$ production under putative Enceladus-like conditions.

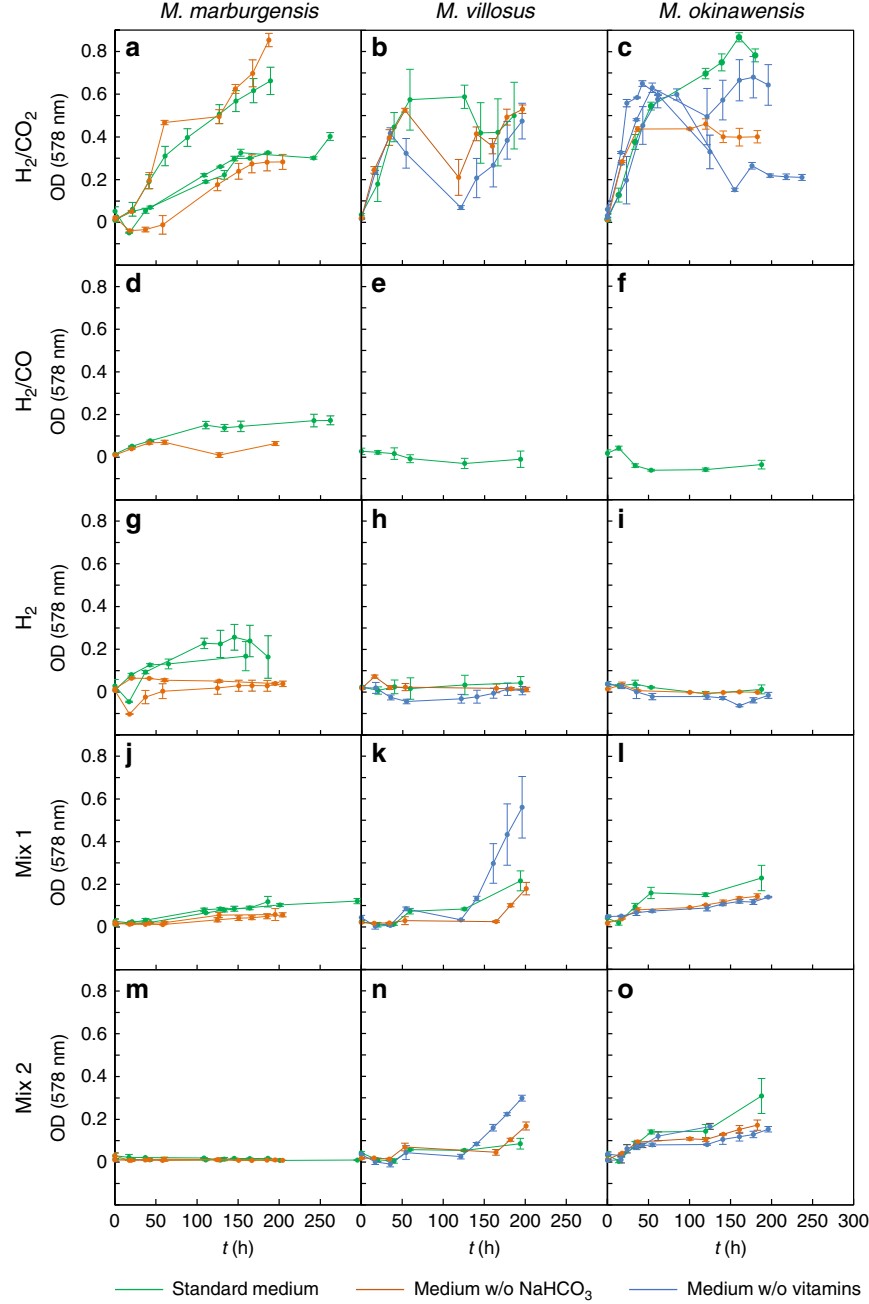

**Fig. 1** Influence of the different headspace gas compositions on growth of *M. marburgensis*, *M. villosus*, and *M. okinawensis*. The error bars show standard deviations calculated from triplicates. OD curves of **a**, **d**, **g**, **j**, **m** *M. marburgensis*, **b**, **e**, **h**, **k**, **n** *M. villosus* and **c**, **f**, **i**, **l**, **o** *M. marburgensis* for **a**–**c** $H_2/CO_2$, **d**–**f** $H_2/CO$, **g**–**i** $H_2$, **j**–**l** Mix 1, and **m**–**o** Mix 2. Growth of *M. marburgensis* was inhibited by the presence of $C_2H_4$ (see Table 2 for detailed gas composition). Only *M. marburgensis* seemed to be able to use sodium hydrogen carbonate (supplied in the medium) as C-source in case of a lack of $CO_2$ ($H_2$ or $H_2/CO$ as sole gas in the headspace). Both, *M. villosus* and *M. okinawensis* showed growth when Mix 1 and Mix 2 were applied to the serum bottle headspace; however, *M. villosus* exhibited extended lag phases. The dips in the graphs **b**, **c** were caused by substrate limitation due to depletion of serum bottle headspace of $H_2/CO_2$ at high-optical cell densities

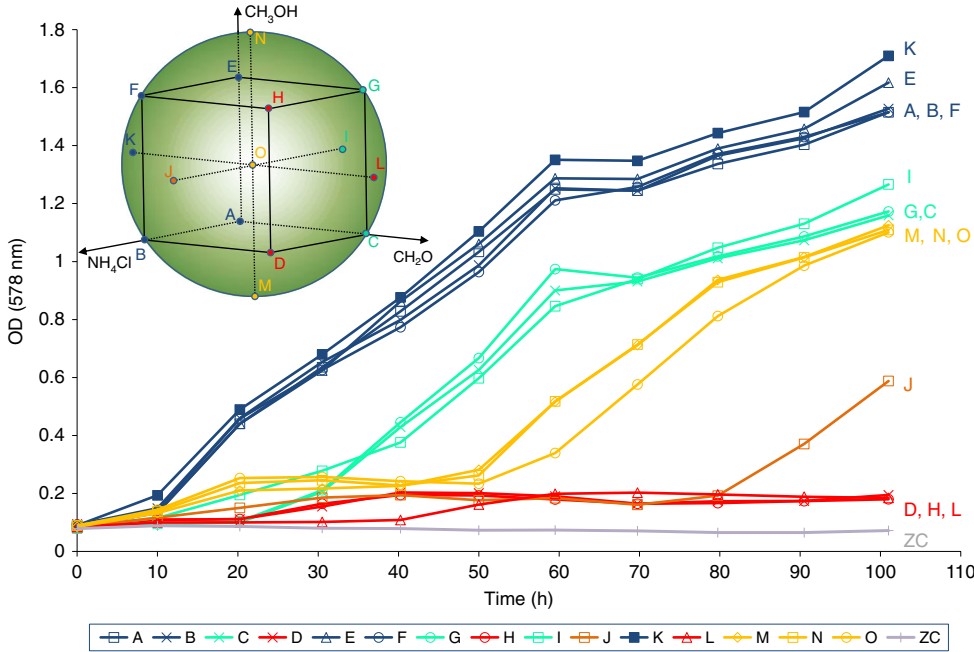

**Fig. 2** Schematic of the experimental setting and DoE raw data growth curves showing OD measurements. The DoE is based on a central composite design (figure in the upper left corner). $NH_4Cl$, $CH_2O$, and $CH_3OH$ were used as factors during the experiment and systematically varied in a multivariate design space (see Supplementary Table 1 for the concrete values). Each of the factors setting was examined in triplicates. The centre point (O) was examined in quintuplicates. The colours of the dots and the letters of the figure in the upper left corner correspond to the growth curves. The line labelled ZC represents the optical density of a corresponding zero control experiment, which was done with the same medium as the experiments labelled with O (central point), but without inoculum. The different colours represent different performances. For better readability, the error bars in this diagram were excluded, which were in a standard deviation range between 0.0009 and 0.1544. According to statistical selection criteria three experiments (one experiment F and two experiments O) were excluded from ANOVA analysis (Supplementary Table 2)

**Table 3 Concentrations of gaseous species in growth medium**

| Gas phase | $P_{H2}$ (bar)[a] | $P_{CO2}$ (bar)[a] | Concentration ($H_2$, mol $L^{-1}$) | Concentration ($CO_2$, mol $L^{-1}$) |
|---|---|---|---|---|
| *120 mL bottles (2 bar)* | | | | |
| $H_2/CO_2$[a] | 2.40 | 0.60 | $1.52 \times 10^{-3}$ | $7.73 \times 10^{-3}$ |
| $H_2/CO$[a] | 2.41 | | $1.52 \times 10^{-3}$ | |
| $H_2$[a] | 3.00 | | $1.89 \times 10^{-3}$ | |
| Mix 1[a] | 0.69 | 0.58 | $4.34 \times 10^{-4}$ | $7.57 \times 10^{-3}$ |
| Mix 2[a] | 0.67 | 0.58 | $4.25 \times 10^{-4}$ | $7.46 \times 10^{-3}$ |
| *0.7 L reactor pressure tests* | | | | |
| $H_2/CO_2$[b] (~10 bar) | 8.00 | 2.00 | $5.05 \times 10^{-3}$ | $2.59 \times 10^{-2}$ |
| $H_2/CO_2$[b] (~20 bar) | 16.00 | 4.00 | $1.01 \times 10^{-2}$ | $5.18 \times 10^{-2}$ |
| $H_2/CO_2$[b] (~50 bar) | 40.00 | 10.00 | $2.53 \times 10^{-2}$ | $1.29 \times 10^{-1}$ |
| $H_2/CO_2/N_2$[c] (~20 bar) | 8.00 | 2.00 | $5.05 \times 10^{-3}$ | $2.59 \times 10^{-2}$ |
| $H_2/CO_2/N_2$[c] (~90 bar) | 36.00 | 9.00 | $2.27 \times 10^{-2}$ | $1.17 \times 10^{-1}$ |
| *2.0 L reactor pressure and inhibitor tests I* | | | | |
| Mix[d] (~10 bar) | 5.60 | 0.70 | $3.53 \times 10^{-3}$ | $9.07 \times 10^{-3}$ |
| Mix[d] (~25 bar) | 10.40 | 1.20 | $6.57 \times 10^{-3}$ | $1.55 \times 10^{-2}$ |
| $H_2/CO_2$[b] (~20 bar) | 16.30 | 4.00 | $1.03 \times 10^{-2}$ | $5.18 \times 10^{-2}$ |
| Mix[d] (~50 bar) | 27.50 | 3.10 | $1.74 \times 10^{-2}$ | $4.01 \times 10^{-2}$ |
| *2.0 L reactor pressure and inhibitor tests II* | | | | |
| Mix[d] (~10 bar) | 5.90 | 0.70 | $3.72 \times 10^{-3}$ | $9.06 \times 10^{-3}$ |
| Mix[d] (~25 bar) | 11.00 | 1.20 | $6.94 \times 10^{-3}$ | $1.55 \times 10^{-2}$ |
| $H_2/CO_2$[b] (~20 bar) | 16.30 | 4.00 | $1.03 \times 10^{-2}$ | $5.18 \times 10^{-2}$ |
| Mix[d] (~50 bar) | 27.70 | 3.30 | $1.75 \times 10^{-2}$ | $4.27 \times 10^{-2}$ |
| *Cassini*[e] | | | | |
| 1 bar | 0.50 | 0.10 | $3.18 \times 10^{-4}$ | $1.31 \times 10^{-3}$ |
| 50 bar | 25.21 | 5.04 | $1.59 \times 10^{-2}$ | $6.53 \times 10^{-2}$ |

[a] Detailed composition of the gases can be found in Table 2
[b] For the $H_2/CO_2$ experiments a ratio of 4:1 was applied
[c] For the $H_2/CO_2/N_2$ experiments a ratio of 4:1:5 was applied
[d] Detailed composition of the gases can be found in Table 4
[e] For the Cassini estimations, a hydrostatic pressure assumed to prevail in the Enceladus' ocean (50 bar) was applied. $H_2$ and $CO_2$ mixing ratio was taken from Table 1

| | N₂ (Vol.-%) | H₂ (Vol.-%) | CO₂ (Vol.-%) | CO (Vol.-%) | C₂H₄ (Vol.-%) |
|---|---|---|---|---|---|
| **Table 4 Gas composition of experiments performed in the 2.0 L bioreactor** | | | | | |
| *2.0 L reactor pressure and inhibitor tests I* | | | | | |
| 20 bar (H₂/CO₂) | | 80.29 | 19.71 | | |
| 10 bar | 31.43 | 53.33 | 6.67 | 4.76 | 3.81 |
| 25 bar | 42.91 | 42.11 | 4.86 | 4.45 | 5.67 |
| 50 bar | 32.67 | 55.11 | 6.21 | 3.01 | 3.01 |
| *2.0 L reactor pressure and inhibitor tests II* | | | | | |
| 20 bar (H₂/CO₂) | | 80.30 | 19.70 | | |
| 10 bar | 29.25 | 55.66 | 6.60 | 4.72 | 3.77 |
| 25 bar | 41.43 | 43.82 | 4.78 | 4.38 | 5.58 |
| 50 bar | 32.41 | 55.07 | 6.56 | 2.98 | 2.98 |

**M. okinawensis tolerates Enceladus-like conditions at 2 bar.** Growth and turnover rates (calculated via the decrease in headspace pressure) of *M. okinawensis* cultures were determined in the presence of selected putative liquid inhibitors detected in Enceladus' plume ($NH_3$, given as $NH_4Cl$, $CH_2O$, and $CH_3OH$). While growth of *M. okinawensis* could still be observed at the highest concentration of $NH_4Cl$ added to the medium (16.25 g L$^{-1}$ or 0.30 mol L$^{-1}$), the organism grew only in the presence of up to 0.28 mL L$^{-1}$ (0.01 mol L$^{-1}$) $CH_2O$. This is less than, but importantly still in the same order of magnitude of, the observed maximum value of 0.343 mL L$^{-1}$ $CH_2O$ detected in the plume[25]. Growth and $CH_4$ production of *M. okinawensis* in closed batch cultivation was shown at $CH_3OH$ and $NH_4Cl$ concentrations exceeding those reported for Enceladus' plume[4,5,25,26].

To explore how the presence of these inhibitors might influence growth and turnover rates of *M. okinawensis*, we have applied these compounds at various concentrations in a multivariate design space setting (Design of Experiment (DoE)). At different concentrations of $CH_2O$, $CH_3OH$, and $NH_4Cl$, *M. okinawensis* cultures showed growth (Fig. 2) and turnover rates from 0.015 ± 0.012 to 0.084 ± 0.018 h$^{-1}$ (Supplementary Fig. 1; experiments L and K in Fig. 2). $CH_3OH$ amendments at concentrations between 9.09 and 210.91 µL L$^{-1}$ (0.22–5.21 mmol L$^{-1}$) did not reduce or improve growth of *M. okinawensis* (Fig. 2 and Supplementary Tables 1 and 2). Compared to the highest applied $CH_2O$ concentration, the turnover rate of *M. okinawensis* was ~5.6-fold higher at the lowest tested concentration. The results of this experiment indicated that *M. okinawensis* possessed a physiological tolerance towards a broad multivariate concentration range of $CH_2O$, $CH_3OH$, and $NH_4Cl$ and was able to perform the autocatalytic conversion of $H_2/CO_2$ to $CH_4$ while gaining energy for growth.

We used the mean liquid inhibitor concentrations for $CH_2O$ determined in the DoE experiment (DoE centre points) and Enceladus-like concentrations for $CH_3OH$ and $NH_4Cl$ (Supplementary Table 3) to test growth and turnover rates of *M. okinawensis*, using different gases in the headspace ($H_2/CO_2$, Mix 1, and Mix 2 (Fig. 3)). Under all tested headspace gas compositions, *M. okinawensis* showed gas-limited growth (max. OD values of 0.67 ± 0.02, 0.17 ± 0.03, and 0.13 ± 0.03 after ~237 h for $H_2/CO_2$, Mix 1 and Mix 2, respectively). The calculated turnover rates correlated with the different convertible amounts of $H_2/CO_2$ in Mix 1 and Mix 2. Hence, *M. okinawensis* was able to grow and to convert $H_2/CO_2$ to $CH_4$ when $CH_2O$, $CH_3OH$, $NH_4Cl$, CO and $C_2H_4$ were present in the growth medium at the concentrations calculated from Cassini's INMS data (assuming 1 bar, compare Tables 1 and 2). The mixing ratios of these putative inhibitors were based on INMS data[4,5,25,26] but higher than those calculated by using the most recent Cassini data[2,6] (Table 1). This

demonstrates that growth and biological $CH_4$ production of *M. okinawensis* is possible even at higher inhibitor concentrations.

**M. okinawensis tolerates Enceladus-like conditions up to 50 bar.** Due to the fact that methanogens on Enceladus would possibly need to grow at hydrostatic pressures up to 80 bar[8] and beyond, the effect of high pressure on the conversion of headspace gas for *M. okinawensis* was examined in a pressure-resistant closed batch bioreactor. Headspace $H_2/CO_2$ conversion and $CH_4$ production was examined at 10, 20, 50, and 90 bar, either using $H_2/CO_2$ in a 4:1 ratio or applying $H_2/CO_2/N_2$ in a 4:1:5 ratio. A gas conversion of >88% was shown for each of the experiments (Supplementary Fig. 2) except for the 90 bar experiment using $H_2/CO_2/N_2$, where the headspace gas conversion was found to be at 66.4%. However, no headspace gas conversion and $CH_4$ production could be detected when cultivating *M. okinawensis* at 90 bar using $H_2/CO_2$ only (data not shown).

Final experiments were designed to investigate headspace $H_2/CO_2$ conversion and $CH_4$ production of *M. okinawensis* according to INMS data (Table 3) and under conditions of high pressure (10.7 ± 0.1, 25.0 ± 0.7, and 50.4 ± 1.7 bar). Turnover rate, methane evolution rate (MER, calculated via pressure drop) and biological $CH_4$ production (calculated via gas chromatography measurements) for these experiments are shown in Fig. 4. When simultaneously applying putative gaseous (Table 4) and liquid inhibitors (Supplementary Table 3) under high-pressure conditions, we reproducibly demonstrated that *M. okinawensis* was able to perform $H_2/CO_2$ conversion and $CH_4$ production under Enceladus-like conditions.

**Methanogenic life could be fuelled by H₂ from serpentinization.** In light of these experimental findings and the presence of $H_2$ in Enceladus' plume[2], the question arose if serpentinization reactions can support a rate of $H_2$ production that is high enough to sustain autotrophic, hydrogenotrophic methanogenic life. To address this question, we used the PHREEQC[27] code to model serpentinization-based $H_2$ production rates under Enceladus-like conditions (Table 5) with the assumption that the rate-limiting step of the serpentinization reaction is the dissolution of olivine. $H_2$ production rates are poorly constrained, as they strongly depend on assumed grain size and temperature. These rates correspond to the low end of the range of $H_2$ production rates, which were based on a thermal cooling and cracking model[28]. Of the many reactions involved in serpentinization of peridotite, dissolution of the Fe(II)-bearing primary phases is a critical one[29], and the only one for which kinetic data are available. In the model, $CO_2$ reduction to $CH_4$ is predicted to take place once enough $H_2$ in the system was produced to generate thermodynamic drive for the reaction. While abiotic $CH_4$ production is kinetically more sluggish than olivine dissolution[30], biological $CH_4$ production is fast and may be controlled by the rate at which $H_2$ is supplied. The abiotic $CH_4$ production rates listed in Table 5 are hence also modelled such that olivine dissolution is the rate-limiting step. The results of these thermodynamic and kinetic computations show that $H_2$ and $CH_4$ production is predicted for a range of rock compositions (Table 5) and temperature conditions (Supplementary Table 4). The model system essentially represents a closed system with high-rock porosity, such as proposed for Enceladus[2]. The computational results predict how much $H_2$ and $CH_4$ should form within the intergranular space inside Enceladus' silicate core with water-to-rock-ratios between 0.09 and 0.12 (Table 5). The serpentinization reactions are predicted to produce solutions with circumneutral to high pH between 7.3 and 11.3, as well as amounts of $H_2$ that greatly exceed the amount of dissolved inorganic carbon (DIC) trapped in the

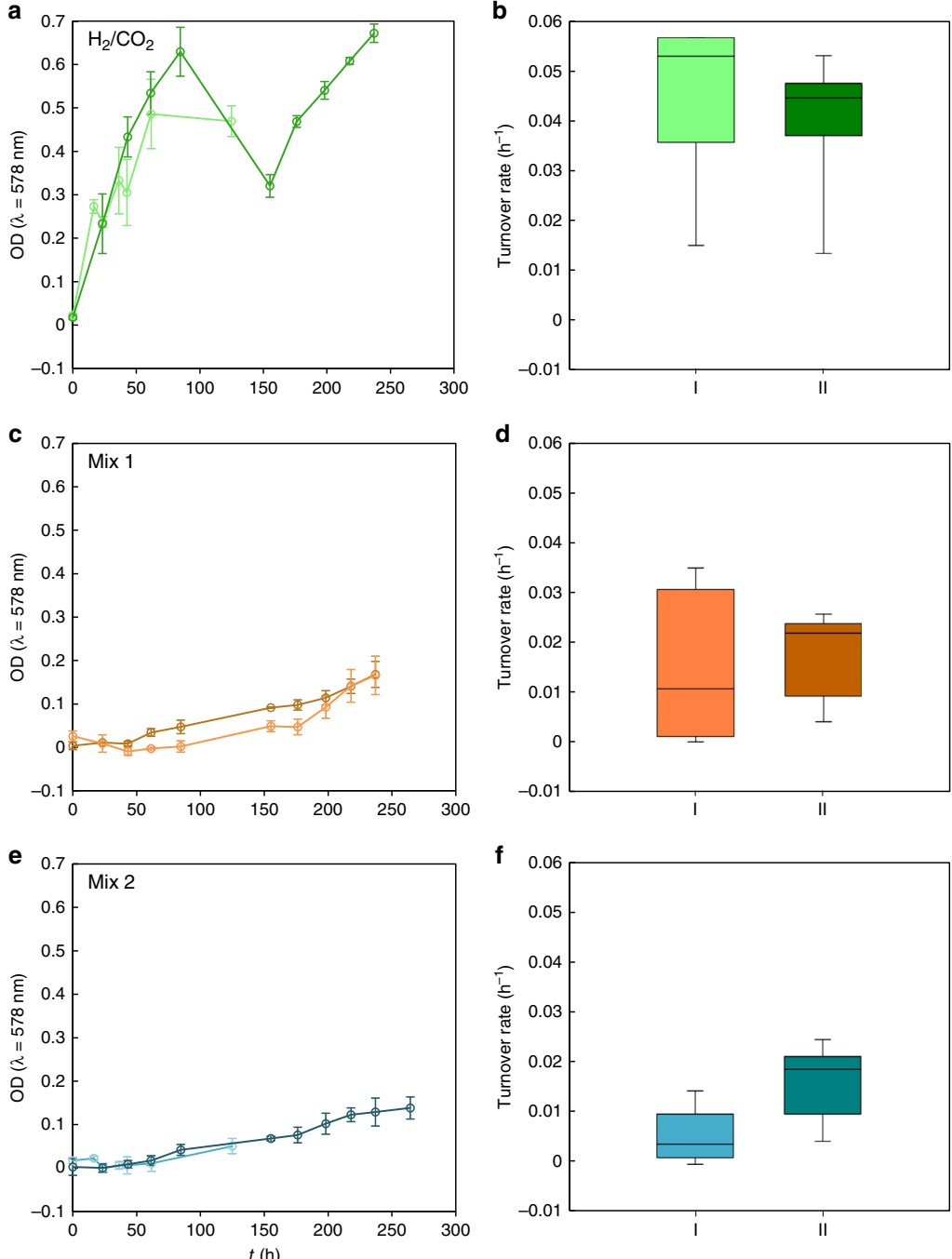

**Fig. 3** Growth and turnover rate of *M. okinawensis* under Enceladus-like conditions at 2 bar. **a**, **c**, **e** Growth curves ($OD_{578\,nm}$) and **b**, **d**, **f** turnover rates ($h^{-1}$) as a measure of $CH_4$ production of *M. okinawensis* on **a**, **b** $H_2/CO_2$ (4:1), **c**, **d** Mix 1 and **e**, **f** Mix 2. For detailed composition of gases and media see Table 2 and Supplementary Table 3. I and II (light and dark colours, respectively) denote two independent experiments (each performed in triplicates, error bars = standard deviation). Enceladus-like concentrations were used for $NH_4Cl$ and $CH_3OH$ and mean liquid inhibitor concentrations determined in the DoE were used for $CH_2O$. The dip in **a** was caused by substrate limitation due to depletion of serum bottle headspace of $H_2/CO_2$ at high-optical cell densities

pore space. As the computations indicate that there is ample thermodynamic drive for reducing DIC to $CH_4$, these results corroborate the idea that serpentinization reactions on Enceladus might fuel autotrophic, hydrogenotrophic methanogenic life. However, we would like to point out that if methanogenic life were indeed active on Enceladus, biological $CH_4$ production would always compete with abiotic $CH_4$ generation processes resulting in a mixed $CH_4$ production.

## Discussion

In this study, we show that the methanogenic strain *M. okinawensis* is able to propagate and/or to produce $CH_4$ under putative Enceladus-like conditions. *M. okinawensis* was cultivated under high-pressure (up to 50 bar) conditions in defined growth medium and gas phase, including several potential inhibitors that were detected in Enceladus' plume[2,4,6]. The only difference between the growth conditions of *M. okinawensis* and the

**Table 5 H$_2$ and CH$_4$ production rates from serpentinization calculated for 50 °C and 50 bar**

| Mineral assemblage[a] | pH | H$_2$ production rate (nmol g$^{-1}$ L$^{-1}$ d$^{-1}$) | CH$_4$ production rate[b] (nmol g$^{-1}$ L$^{-1}$ d$^{-1}$) | Water: rock ratio | Mol H$_2$ produced per mol olivine |
|---|---|---|---|---|---|
| Fo$_{90}$:En:Diop = 8:1:1 | 11.3 | 4.58 | 2.05 | 0.126 | 0.002 |
| Fo$_{90}$ | 8.60 | 4.03 | 1.07 | 0.124 | 0.004 |
| Fo$_{50}$ | 7.50 | 34.7 | 1.35 | 0.104 | 0.033 |
| Fo$_{20}$ | 7.29 | 50.7 | 1.32 | 0.094 | 0.053 |

[a] Fo$_{90}$ = forsteritic olivine (forsterite:fayalaite = 9:1), En = enstatite, Diop = diopside, Fo$_{50}$ = (Fo:Fa = 1:1), Fo$_{20}$ = (Fo:Fa = 2:8)
[b] CH$_4$ production is predicted from allowing H$_2$ and dissolved inorganic carbon (DIC) to equilibrate readily, while H$_2$ production is kinetically controlled by dissolution of primary phases

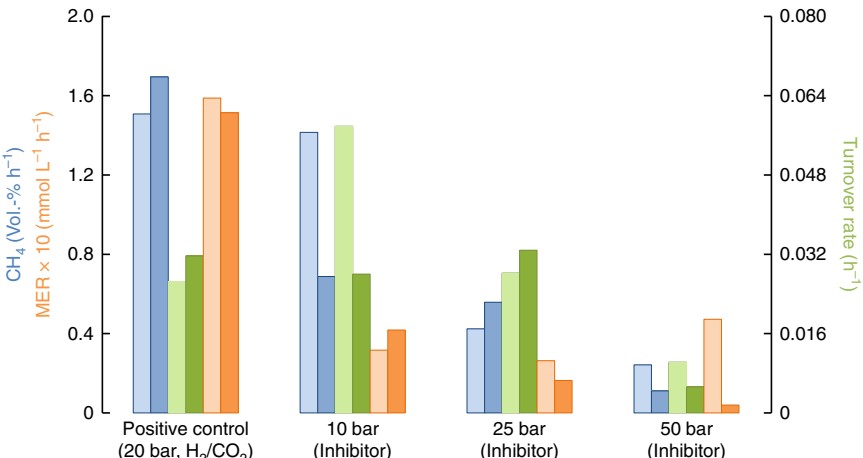

**Fig. 4** CH$_4$ production, MER, and turnover rate of *M. okinawensis* under Enceladus-like conditions at high pressure. Biological CH$_4$ production determined by gas chromatography (blue) (Vol.-% h$^{-1}$) and turnover rates (h$^{-1}$) (green) and MER·10 (mmol L$^{-1}$ h$^{-1}$) (red) measured from headspace gas conversion using *M. okinawensis* (experiment 1 in light colours, experiment 2 in dark colours) under putative Enceladus-like conditions in a 2.0 L bioreactor (for detailed medium composition, see Supplementary Table 3 and for detailed gas composition see Table 4, n = 2). The positive control experiment contained also the liquid inhibitors but only H$_2$/CO$_2$ (4:1) in the headspace. For high-pressure experiments without any inhibitors see Supplementary Fig. 2

putative Enceladus-like conditions was the lower pH value applied during the high-pressure experiments. Due to the supply of CO$_2$ at high-pressure in the experiments the pH decreased to ~5, while pH values between 7.3 and 13.5 were estimated for Enceladus' subsurface ocean (this study and refs. [8,14]).

Another point of debate might be the cultivation temperatures used for the thermophilic and hyperthermophilic methanogens in this study. The mean temperature in the subsurface ocean of Enceladus might be just above 0 °C except for the areas where hydrothermal activity is assumed to occur. In these hydrothermal settings temperatures higher than 90 °C are supposedly possible[8], and are therefore the most likely sites for higher biological activity on Enceladus. Although methanogens are found over a wide temperature range on Earth, including temperatures around 0 °C[31], growth of these organisms at low temperatures is observed to be slow[13].

We estimated H$_2$ production rates between 4.03 and 50.7 nmol g$^{-1}$ L$^{-1}$ d$^{-1}$ in the course of serpentinization on Enceladus (Table 5). These estimates are rather conservative, as they are based on the assumption of small specific mineral surface areas. In a recent study, the rate of serpentinization has been estimated from a physical model that predicts how fast cracking fronts propagate down into Enceladus' core[28]. Combining this physically controlled advancement of serpentinization ($8 \times 10^{11}$ g y$^{[-128]}$) with our estimates for kinetically limited rates of H$_2$ production leads to overall rates of $3-40 \times 10^4$ mol H$_2$ y$^{-1}$ for Enceladus. Although still high enough to support biological methanogenesis, these rates are orders of magnitude lower than the previously suggested $10 \times 10^8$ mol H$_2$ y$^{[-128]}$ assuming that the speed of cracking front propagation controls the rate of H$_2$

production. We hence suggest that reaction kinetics may play an important role in determining the overall H$_2$ production rate on Enceladus. Our computed steady-state H$_2$ production rates are lower than the $1-5 \times 10^9$ mol H$_2$ y$^{-1}$ estimated from Cassini data[2]. This apparent discrepancy in flux rates can be reconciled if the Enceladus plume was a transient (i.e., non-steady state) phenomenon. The predicted H$_2$/CH$_4$ ratio of 2.5 (Fo$_{90}$:En:Diop = 8:1:1) to 4 (Fo$_{90}$) for the magnesian compositions of Enceladus' core (Table 5) are consistent with the relative proportions of the two gases in the plume (0.4–1.4% H$_2$, 0.1–0.3% CH$_4$)[2].

Based on our estimated H$_2$ production rate, we can calculate how much of the available DIC on Enceladus could be fixed into biomass through autotrophic, hydrogenotrophic methanogenesis. If we assume a typical elementary composition of methanogen biomass[32], 7.13 g carbon could be fixed per g hydrogen fixed. Under optimal growth conditions, ~3%[22,23] of the available carbon can be assimilated into biomass, and assuming that methanogens possess a molecular weight of ~30.97 g C-mol$^{[-122]}$ and that the total amount of H$_2$ produced would be available for the carbon and energy metabolism of autotrophic, hydrogenotrophic methanogens, a biomass production rate between 20 and 257 C-nmol g$^{-1}$ L$^{-1}$ y$^{-1}$ could be achieved. In another approach, we can use the actually predicted CH$_4$ production rates of 1.32–2.05 nmol g$^{-1}$ L$^{-1}$ d$^{-1}$ (Table 5) and a Gibbs energy dissipation approach in which we assume that 10% of the energy of CH$_4$ production is fuelling biosynthesis[28]. This yields similar numbers of biomass production rate, i.e., 28 and 56 C-nmol g$^{-1}$ L$^{-1}$ y$^{-1}$.

Based on our findings, it might be interesting to search for methanogenic biosignatures on icy moons in future space

missions. Methanogens produce distinct and lasting biosignatures, in particular lipid biomarkers like ether lipids and isoprenoid hydrocarbons. Other potential biomarkers for methanogens are high-nickel (Ni) concentrations (and its stable isotopes[33]), as Ni is e.g., part of methyl-coenzyme M reductase, the key enzyme of biological methanogenesis[23]. However, both lipid biomarkers and Ni-based biosignatures are likely only to be identifiable at the site of biological methanogenesis, and the effect of dilution with increasing distance away from the methanogen habitat is likely to prevent their use as a general marker for biological methanogenesis in Enceladus' plume or in a subsurface ocean. If, however, bubble scrubbing would occur, a process by which organic compounds and cells adhere to bubble surfaces and are carried away as bubbles rise, which was suggested to occur on Enceladus[34], the amount of bioorganic molecules and cells would be much higher and future lander missions could easily collect physical evidence for the presence of autotrophic, hydrogenotrophic methanogenic life on Enceladus.

Additionally, one could consider using stable isotopes of $CH_4$ and $CO_2$ and ratios of low-molecular weight hydrocarbons to evaluate the possibility of biological methanogenesis on Enceladus[11]. But given the uncertainties on the geological and hydrogeological boundary conditions that influence the targeted isotope and molecular patterns in Enceladus' plume, such an approach is not trivial. In contrast to biological and thermogenic $CH_4$ production, the latter resulting from the decomposition of organic matter, abiogenic $CH_4$ is believed to be produced by metal-catalysed Fischer–Tropsch or Sabatier type reactions under hydrothermal conditions and particularly in the course of serpentinization of ultramafic rocks[35]. Although biologically produced $CH_4$ is usually characterised by its strong $^{13}C$ depletion, growth of methanogens at high-hydrostatic pressures and high temperatures, which is typical of deep-sea hydrothermal systems, may significantly reduce kinetic isotope fractionation and result in relatively high $\delta^{13}C$ values of $CH_4$, hampering discrimination from non-microbial $CH_4$[36]. Given such uncertainties, multiply substituted, so-called 'clumped' isotopologues of $CH_4$ emerge as new proxy to constrain its mode of formation and to recognise formation environments like serpentinization sites[37].

Another approach to identify the origin of $CH_4$ could be $CH_4$/(ethane + propane) ratios, as low ratios are typical of settings dominated by thermogenic $CH_4$[38]. However, this ratio may fall short to unequivocally discriminate abiogenic from biologically produced $CH_4$. For instance the ratio of $CH_4$ concentration to the sum of $C_{2+}$ hydrocarbon concentration ($C_1/C_{2+}$) in the serpentinite-hosted Lost City Hydrothermal Field of $950 \pm 76$ was found to be most similar to ratios obtained in experiments with Fischer–Tropsch type reactions (<100–>3000). Thermogenic reactions produce $C_1/C_{2+}$ ratios less than ~100, whereas biological methanogenesis results in ratios of 2000–13000[39]. More than 30 years of research on $CH_4$ production have revealed that its biologic, thermogenic or abiogenic origin on Earth is often difficult to trace[40]. However, the experimental and modelling results presented in this study together with the estimates of the physicochemical conditions on Enceladus from earlier contributions make it worthwhile to increase efforts in the search for signatures for autotrophic, hydrogenotrophic methanogenic life on Enceladus and beyond.

## Methods

**Estimations of Enceladus' interior structure**. Due to its rather small radius, the uncompressed density of the satellite is almost equal to its bulk density, which makes a simplified model of Enceladus' interior reasonable. Enceladus was divided into a rocky core (core density of 2300–2550 kg m$^{-3}$), a liquid water layer (density of 960–1080 kg m$^{-3}$), and an icy shell (ice density of 850–960 kg m$^{-3}$) and hydrostatic equilibrium was assumed. Calculations of the hydrostatic pressure based on Enceladus mass of $1.0794 \times 10^{20}$ kg[41] and its mean radius of 252.1 km[41]

assuming a core radius of 190–200 km, an subsurface ocean depth of 60–10 km and a corresponding ice shell thickness of 2.1–52.1 km results in a pressure of ~44.3–25.2 bar or 80.1–56.2 bar (depending on the method, Supplementary Methods) at the water-core-boundary. For the high-pressure experiments in this study, including all inhibitors, three pressure values were chosen that lie in the range given by Hsu et al. (10–80 bar)[8] and are related to our calculations, i.e., 10, 25, and 50 bar.

**Low-pressure experiments**. Growth and tolerance towards putative inhibitors of the three methanogenic strains *Methanothermococcus okinawensis* DSM 14208, *Methanothermobacter marburgensis* DSM 2133, and *Methanococcus villosus* DSM 22612 were elucidated (Fig. 1). All strains were obtained from the Deutsche Stammsammlung von Mikroorganismen und Zellkulturen GmbH, Braunschweig, Germany. Growth was resolved by optical density (OD) measurements ($\lambda = 578$ nm). $H_2/CO_2$–$CH_4$ gas conversion [%], turnover rate [h$^{-1}$] (see Equation (2) below), and MER were calculated from the decrease of the bottle headspace pressure and/or from measuring $CH_4$ production in a closed batch setup[22,24]. The headspace pressures were measured using a digital manometer (LEO1-Ei, −1…3barrel, Keller, Jestetten, Germany) with filters (sterile syringe filters, w/0.2c μm cellulose, 514-0061, VWR International, Vienna, Austria), and cannulas (Gr 14, $0.60 \times 30$ mm, 23 G × 1 1/4", RX129.1, Braun, Maria Enzersdorf, Austria). The detailed setting can be seen in Fig. 3(a) in Taubner and Rittmann[22]. All pressure values presented in this study are indicated as relative pressure in bar.

For the experiments at 2 bar regarding CO and $C_2H_4$ tolerance (see Figs. 1 and 3), the strains were incubated in the dark either in a water bath (*M. marburgensis* and *M. okinawensis*, $65 \pm 1$ °C) or in an air bath (*M. villosus*, $80 \pm 1$ °C). The methanogens were cultivated in 50 mL of their respective chemically defined growth medium. Compositions of the different growth media of the experiments shown in Figs. 1 and 3 can be found in Supplementary Tables 3 and 5–10. The final preparation of the medium in the anaerobic culture flasks was performed in an anaerobic chamber (Coy Laboratory Products, Grass Lake, USA). Experiments were performed over a time of 210–270 h. After each incubation period, serum bottle headspace pressure measurement (in order to be unbiased, flasks were previously cooled down to room temperature), OD-sampling, and gassing with designated gas or test gas was performed. OD measurement was performed at 578 nm in a spectrophotometer (DU800, Beckman Coulter, USA). A zero control was incubated together with each individual experiment and the OD of this control was subtracted from the measured OD from the inoculated flasks each time.

For hydrogenotrophic methanogens, which utilise $H_2$ as electron donor for the reduction of $CO_2$ to produce $CH_4$ and $H_2O$ as their metabolic products, the following stoichiometric reaction equation was used[12,23,24]:

$$4H_{2(g)} + CO_{2(g)} \rightarrow CH_{4(g)} + 2H_2O_{(aq)} \quad \Delta G^0 = -135 \text{ kJ mol}^{-1}. \quad (1)$$

The turnover rate [h$^{-1}$] correlates with the catalytic efficiency per unit of time, i.e., it is a way to indirectly quantify $CH_4$ productivity. By assuming the above-mentioned chemical $CO_2$ methanation stoichiometry and neglecting biomass formation, the turnover rate is an equivalent method for indirect quantification of $CH_4$ production. It is defined as

$$\text{turnover rate [h}^{-1}\text{]} = \frac{\Delta p}{\Delta p_{max} \cdot \Delta t}, \quad (2)$$

where $\Delta p$ [bar] is the difference in pressure before and after incubation, $\Delta p_{max}$ [bar] is the maximal theoretical difference that would be feasible due to stoichiometric reasons[22], and $\Delta t$ [h] is the time period of incubation.

For the initial pressure experiments at 2 bar, the three methanogenic strains were tested under five different gas phase compositions (Table 2). A significant change in OD and turnover rate was observed between these experiments (as can be seen in Fig. 1). When Mix 1 and Mix 2 were applied, only a maximum of $22.66 \pm 0.23$ Vol.-% $H_2$ (average, Table 2) could be converted to $CH_4$ and biomass.

To evaluate the influences of the potential inhibitors $NH_3$, $CH_2O$, and $CH_3OH$ on the growth of *M. okinawensis*, several preliminary experiments were performed. For easier handling, $NH_3$ was substituted by $NH_4Cl$. Based on INMS data (Table 1) the amount of $NH_4Cl$ was calculated according to Henry's law. For that, Henry's law constant was calculated to be 0.1084 mol m$^{-3}$ Pa$^{-1}$ at 64 °C. This results in 11.6 g L$^{-1}$ (0.22 mol L$^{-1}$) $NH_4Cl$ to have ~1% of $NH_3$ in the gaseous phase at equilibrium for the experiments under closed batch conditions. The influence of $NH_4Cl$ between 0.25 and 16.25 g L$^{-1}$ (4.67 and 303.79 mmol L$^{-1}$), $CH_2O$ between 0 to 111 μL L$^{-1}$ (0–4.03 mmol L$^{-1}$), and $CH_3OH$ between 0 and 200 μL L$^{-1}$ (0–4.94 mmol L$^{-1}$) was tested individually. $CH_2O$ (37 Vol.-%) and $CH_3OH$ (98 Vol.-%) were used as stock solutions.

To find an appropriate ratio for the final experiments, an experiment in a DoE setting was established. A central composite design with the parameters shown in Supplementary Table 1 and Fig. 2 was chosen. The design space is spherical with a normalised radius equal to one. Experiments A–N were done in triplicates; experiments O were performed in quintuplicate. The results of these experiments in terms of OD can be seen in Fig. 2 and in terms of turnover rate in

Supplementary Fig. 1. Each incubation time period was $10.0 \pm 0.5$ h. The ANOVA analysis of this study can be found in Supplementary Table 2.

The setting for the experiments under Enceladus-like conditions at 2 bar pressure included the medium described in Supplementary Table 3 and Mix 2 (Table 2) as gaseous phase. As can be seen in Fig. 3 there was a lag phase of two days, but after that continuous but slow growth was observed.

To calculate the molar concentration of $H_2$ and $CO_2$ in the medium (Table 3), Henry's law was used:

$$M = k_H \cdot p_X, \tag{3}$$

where $p_X$ is the partial pressure of the respective gas and $k_H$ is Henry's constant as a function of temperature:

$$k_H = k_H^{\ominus} \cdot e^{\left(\frac{-\Delta_{soln}H}{R}\left(\frac{1}{T} - \frac{1}{T^{\ominus}}\right)\right)}, \tag{4}$$

where $\Delta_{soln}H$ is the enthalpy change of the dissolution reaction. For $k_H^{\ominus}$ the values $7.9 \times 10^{-4}$ mol $L^{-1}$ $bar^{-1}$ and $3.4 \times 10^{-2}$ mol $L^{-1}$ $bar^{-1}$ and for $\frac{-\Delta_{soln}H}{R}$ the values 500 K and 2400 K for $H_2$ and $CO_2$, respectively, were used. This results in a Henry constant at 65 °C of $6.481 \times 10^{-4}$ mol $L^{-1}$ $bar^{-1}$ and $1.329 \times 10^{-2}$ mol $L^{-1}$ $bar^{-1}$ for $H_2$ and $CO_2$, respectively.

Another potential liquid inhibitor detected in Enceladus' plume was hydrogen cyanide (HCN)[3,4]. However, calculations on HCN stability under the assumed conditions on Enceladus show that HCN would hydrolyse into formic acid and ammonia[42]. Further investigations on the stability of HCN at different pH values and temperatures yielded similar results[43,44]. It was therefore assumed that HCN might originate either from a very young pool, a recent aqueous melt, or from the icy matrix on Enceladus[4,42]. Due to this reasoning and the low probability of HCN presence in the subsurface ocean of Enceladus, HCN was neglected in all growth media used to perform the experiments.

**High-pressure experiments**. *M. okinawensis* initial high-pressure experiments were performed at its optimal growth temperature of $65 \pm 1$ °C using a chemically defined medium (250 mL, see Supplementary Table 10 for exact composition) and a fixed stirrer speed of 100 r.p.m. in a 0.7 L stirred stainless steel Büchi reactor. Before each of the experiments, the reactor was filled with medium and the entire setting was autoclaved under $CO_2$ atmosphere to assure sterile conditions. Thereafter, the inoculum (1 Vol.-%), the $NaHCO_3$, L-cysteine, $Na_2S \cdot 9 H_2O$ (0.5 M) and trace element solution were transferred via a previously autoclaved transfer vessel into the reactor. Then the reactor was set under pressure with the selected gas mixture (added ~5 bar in discrete steps every 10 min). The initial high-pressure experiments were performed using both an $H_2/CO_2$ (4:1) gas phase and an $H_2/CO_2/N_2$ (4:1:5) mixture. The reactor was equipped with an online pressure sensor (ASIC Performer pressure sensor 0–400 bar, Parker Hannifin Corporation, USA) and temperature probe (thermo element PT100, −75 °C − 350 °C, TC Mess- und Regeltechnik GmbH, Mönchengladbach, Germany). The conversion was always above 88% except for the 90 bar experiment, wherein also $N_2$ fixation into biomass could be assumed. Interestingly, the time until start of the conversion decreases for the $H_2/CO_2$ experiments upon an increase of headspace pressure in the initial setup. This could be an indication for a barophilic nature of this organism, but also due to the experimental closed batch setup.

Increasing $p_{CO_2}$ and associated pH change was determined using a pH probe (see Supplementary Fig. 3). This analysis showed that even a rather small $p_{CO_2}$ (2 bar) already decreases the pH from nearly neutral to >5 due to the medium composition, which is due to application of a medium with low-buffering capacity, also possibly occurring in Enceladus' subsurface ocean. However, it remains an open question if the medium on Enceladus is buffered. This would lead to higher possible $p_{CO_2}$[45] without having a drastic influence on the reported pH. Furthermore, we calculated if $NaHCO_3$ could be used as source of dissolved inorganic carbon and what would be the effect on the pH of the medium. Postberg et al. suggested a concentration of 0.02–0.1 mol $kg^{-1}$ $NaCO_3$ and 0.05–0.2 mol $kg^{-1}$ NaCl in the medium to reach a pH level between 8.5 and 9[46]. Calculations on the concentrations of dissolved $CO_2$ in the high-pressure experiments were performed by using the mole fraction of dissolved $CO_2$ in $H_2O$ depending on $p_{CO_2}$. The mole fractions for $p_{CO_2}$ of 0.7, 1.2, and 3.1 bar (as used in the experiments) at 65 °C were generated by extrapolation of given values in the region of $p_{CO_2} = 0$–1 bar. $H_2/CO_2$ conversion and $CH_4$ production could still be measured at 10 bar, 20 bar and 50 (i.e., $p_{CO_2} = 10$) bar with $H_2/CO_2$ (4:1) gas phase, but no decrease in pressure was observed at 90 bar with $H_2/CO_2$ (4:1) gas phase (i.e., $p_{CO_2} = 18$ bar) after >110 h (data not shown). It is assumed that no growth occurs under these conditions due to the high $p_{CO_2}$ (18 bar) and the associated decrease to a pH of <3, which is beyond the reported pH tolerance of *M. okinawensis*[18].

The high-pressure experiments under Enceladus-like conditions were carried out in the presence of both gaseous and liquid inhibitors applying the optimal growth temperature of $65 \pm 1$ °C at individual pressures of 10, 25, and 50 bar in a stirred 2.0 L Büchi reactor at 250 r.p.m. The gas ratios of the final gas mixtures are reported in Table 4. The final liquid medium was the same as the one used in the final 2 bar experiments (incl. the liquid inhibitors, Supplementary Table 3). Due to the results from the pH experiments (Supplementary Fig. 3), the low $p_{CO_2}$ (~3 bar) was chosen to avoid a pH shift to more acidic values, which is not representative of Enceladus-like conditions. For final high-pressure experiments, the medium

volume was set to 1.1 L and *M. okinawensis* $H_2/CO_2$ grown pre cultures of an OD = 34.4 and OD = 34.5 were used as inoculums (11 mL each). Preparation of the high OD *M. okinawensis* suspension was performed by collecting 1 L of serum bottle grown fresh culture (OD ~0.7), centrifuging the cells at 5346×g anaerobically for 20 min (Heraeus Multifuge 4KR Centrifuge, Thermo Fisher Scientific, Osterode, Germany), and re-suspending the cells in 20 mL of freshly reduced appropriate growth medium. The gases were added into the bioreactor headspace in the following order: CO, $N_2$, $CO_2$, $H_2$, and $C_2H_4$ (5 bar every 10 min). During all high-pressure experiments OD measurements were not conducted because upon reactor depressurisation cell envelopes of *M. okinawensis* were found to be disrupted. To determine the amount of produced $CH_4$ in the high-pressure experiments, gas samples were taken after reducing the pressure in the reactor down to $1.36 \pm 0.25$ bar. The sample was stored in 120 mL serum bottles and sealed with black septa (3.0 mm, Butyl/PTFE, La Pha Pack, Langerwehe, Germany). The volumetric concentrations of $CH_4$ were determined using a gas chromatograph (7890 A GC System, Agilent Technologies, Santa Clara, USA) equipped with a TCD detector and a 19808 ShinCarbon ST Micropacked Column (Restek GmbH, Bad Homburg, Germany)[22].

To determine the $CH_4$ production [Vol.-% $h^{-1}$] shown in Fig. 4, the value of $CH_4$ Vol.-% was divided by the time of biological $CH_4$ production in h. To exclude a potential lag phase, the starting point of biological $CH_4$ production was set to the point in time when the decrease in pressure exceeded the initial pressure by 5% for 10 bar experiments or by 1% during the other experiments.

**Serpentinization simulations**. The PHREEQC[27] code was used to simulate serpentinization reactions from 25 to 100 °C and from 25 to 50 bar in order to assess $H_2$ production on Enceladus. The Amm.dat and llnl.dat databases were used for all simulations, which account for temperature and pressure dependent equilibrium constants for dissolved species and solid phases up to 100 °C and 1000 bar. Solution composition was taken from the chemical composition of erupting plume of Enceladus[4], as the true chemistry of its subsurface sea is unknown. Dissolved concentrations of $Ca^{2+}$, $Fe^{2+}$, $Mg^{2+}$, and $SiO_2$ were assumed to be seawater-like and values from McCollom and Bach[47] were used. At the very low water-to-rock ratios of our model, the compositions of the interacting fluids will be entirely rock buffered, so that the model results are insensitive to the choice of the starting fluid composition. DIC concentration was set to 0.04 mol $L^{-1}$ taken from Glein et al.[14], who estimated a possible range of 0.005–1.2 molal DIC in Enceladus' subsurface ocean. The solid phase assemblage was composed of varying amounts of olivine, enstatite and diopside, as well as varying olivine compositions. Most planetary bodies exhibit olivine solid solutions (Mg, Fe)$_2$SiO$_4$ that are dominated by forsterite (Mg$_2$SiO$_4$). This has been shown for micrometeorites found on Earth, lunar meteorites, comets, and asteroids[48–51]. Stony iron meteorites, such as pallasites contain forsteritic olivine with up to 20% $Fe^{2+}$ content[52]. Olivines in chondrites show a more varied compositions ranging between 7 and 70% ferrous iron[53,54], to almost pure fayalite (Fe$_2$SiO$_4$)[55]. A realistic assumption is that olivines on Enceladus have a more forsteritic composition that resembles those of stony iron and lunar meteorites.

A composition of Fo$_{90}$ was adopted for olivine. Calculations were limited to Fo$_{90}$, fayalite, enstatite, and diopside, as experimental data on their dissolution kinetics at high pH and low temperature are available. Kinetic rate laws were applied for forsteritic olivine, fayalite, enstatite, and diopside from Wogelius and Walther[56,57], Daval et al.[58], Oelkers and Schott[59], and Knauss et al.[60], respectively. Dissolution rate laws for Fo$_{90}$ and fayalite at 100 °C were extrapolated from rate data in Wogelius and Walther[57] using the Arrhenius equation. Enstatite and diopside rate laws at 100 °C were power-law fitted from experimental data provided by Oelkers and Schott[59] and Knauss et al.[60], respectively. All rate laws are valid over a pH range from 2 to 12 at all temperatures. Kinetic rate laws were multiplied by the total surface area of each mineral present in solution in order to calculate moles of minerals dissolved per time. Surface areas of 590 cm$^2$ $g^{-1}$ for Fo$_{90}$ and fayalite[61], 800 cm$^2$ $g^{-1}$ for enstatite[59], and 550 cm$^2$ $g^{-1}$ for diopside[60] were used. These specific surface areas have been suggested to be typical for fine-grained terrestrial rocks. We adopted these numbers in our computations, as we have no constraints on what specific surface areas in Enceladus may be. If the core of Enceladus was similar to carbonaceous chondrite, then the average mineral grain size is smaller and hence the specific surface areas greater than we assumed[62]. We choose to use fairly small specific surface areas to provide conservative estimates for $H_2$ production rates. Model 1 uses Fo$_{90}$, enstatite and diopside in a ratio of 8:1:1, model 2 uses a pure Fo$_{90}$ composition. The effect of ferrous iron content in olivine on $H_2$ production rates was tested in computations where Fo$_{50}$ (Fo:Fa 1:1, model 3) and Fo$_{20}$ (Fo:Fa 2:8, model 4) were dissolved as the sole mineral. Fo$_{50}$ and Fo$_{20}$ were dissolved according to dissolution rates of Wogelius and Walther (their equation (6))[57], and for temperatures beyond 25 °C dissolution rates for fayalite were extrapolated to 50 and 100 °C after Daval et al.[58]. Model 1 contained 40 mol of Fo$_{90}$ and 5 mol of enstatite and diopside. For models 2–4, 55 mol of Fo$_{90}$, Fo$_{50}$, and Fo$_{20}$ were used. Applying these amounts yield water-to-rock-ratios between 0.09 and 0.12 (Table 5).

The most likely environmental conditions present within Enceladus are temperatures between 25 and higher than 90 °C at 25–80 bar[8], and temperatures of 50 °C and pressures of 50 bar were chosen for the four different models. In a separate set of computations, temperatures were altered to 25 and 100 °C, and pressures were set at 25 and 100 bar. These results are shown in Supplementary

Table 4. As pressure has a negligible effect on H$_2$ production, only the variations in temperature change are shown.

**Data availability.** The data sets analysed during the current study are available in this article and its Supplementary Information file, or from the corresponding author on request.

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

## Acknowledgements

Barbara Reischl, MSc and Annalisa Abdel Azim, MSc are gratefully acknowledged for expert technical assistance during the closed batch experiments. We thank Dr. Jessica Koslowski for proofreading and comments on the manuscript. R.-S.T. would like to thank Mark Perry and Britney Schmidt for discussions. We also thank David Parkhust for helping with the PHREEQC code and Silas Boye Nissen for his support in the initial attempts of the serpentinization modelling. Financial support was obtained from the Österreichische Forschungsförderungsgesellschaft (FFG) with the Klimafonds Energieforschungsprogramm in the frame of the BioHyMe project (grant 853615). R.-S.T. was financed by the University of Vienna (FPF-234) and a fellowship of L'Oréal Österreich.

## Author contributions

R.-S.T., P.P., C.Pr., P.K., and S.K.-M.R.R. performed the experiments. R.-S.T., P.P., S.B., A.H.S., C.Pa., and S.K.-M.R.R. designed the experiments. D.S. and J.Z. designed and performed PHREEQC modelling. W.B. supervised the PHREEQC modelling. R.-S.T., P.P., J.Z., D.S., S.B., A.H.S., A.K., W.B., J.P., C.Pa, M.G.F., C.S., and S.K.-M.R.R. discussed the data. R.-S.T., J.Z., D.S., W.B., J.P., C.S., and S.K.-M.R.R. wrote the manuscript.
