## [Peer Review File · Nature Communications]

Reviewers' comments:

Reviewer #1 (Remarks to the Author):

The manuscript by Taubner, et al., "Biological methane production under Enceladus-like conditions" describes the results of a series of experiments aimed to measure the growth of autotrophic, hydrogenotrophic methanogens under conditions analogous to those predicted for the icy moon. The authors explore growth rates of 3 methanogenic species in the presence of different gas phases, chemistries, and even elevated hydrostatic pressures- that are expected to occur at the water/rock interface on Enceladus. The work considers the consequences of these variables in their ability to both support or inhibit the growth of the methanogenic species. The topic of habitability of Icy (or "Ocean") Worlds of the outer solar system is extremely timely, building upon recent observational data and models of Enceladus' interior and the contents of plumes emitted through cryovolcanism into space. I applaud the authors' creative attempts to cultivate relevant model microorganisms under non-standard conditions such as may be expected on Enceladus. This is not often done, especially with challenging to grow strains and experimental rigor- both of which were tackled in the current study. From my reading of this manuscript, the major claims of this paper are that 1) methanogenic organisms can function and produce methane under gas compositions and pressures expected to occur on Enceladus and 2) that potentially inhibitory compounds such as formaldehyde, ammonia, carbon monoxide, etc. do not severely limit growth at the concentrations tested. The data presented in this work appears to be thorough and replicable, and provides insight, at the least, on the physiology of methanogenic Archaea.

From my perspective, the work lacked several key elements that would make it important and interesting to a broad audience, beyond the archaeal physiology community. First, it is becoming more and more well-recognized that methane originating from serpentinizing environments on Earth is abiogenic. See recent works by David Wang (Science, 2015) and Giuseppe Etiope (e.g. Applied Geochemistry, 2017). While this does not exclude the contributions of biological methanogenesis to systems on either the modern-day Earth or Enceladus- the complete absence of a mention of biogenic versus abiogenic reaction mechanisms in the current manuscript is noticeable. Why might biogenic processes be more favorable or feasible on Enceladus than abiotic processes? How do known rates and pathways of biotic and processes compare in the physical/chemical environment visualized. Second, I think that the authors missed a major opportunity to discuss how the growth of methanogens under Enceladus-like conditions could contribute to detectable biosignatures in the plumes. How many cells per liter could be supported (similar to the work by Chyba and Gaidos in the 1990's for Europa)? How do the activities of methanogens impact the composition of aqueous and gaseous metabolites? Most importantly, in my opinion there should absolutely be a discussion of stable isotope signatures that have become a key point of focus for terrestrial systems with overlapping biogenic and abiogenic signatures. How might biological activities enhance or suppress these signatures. I am thinking specifically of the work by Ken Takai, et al (2008) studying hyperthermophilic methanogens at extremely high temperatures and pressures, showing a decrease in stable isotope fractionation. However, recent works by Wang, Sherwood Lollar, Etiope, and many others are equally relevant to the discussion. In summary, the current manuscript is interesting and imaginative, and the data appears to be of high quality, thorough, and replicable. However, it also feels as if it only half-way met its potential, without a mention of how our interpretation of methane sources on Earth has been confounding, let alone in the outer solar system.

Specific edits and corrections are mentioned below:

Line 80: Were the growth kinetics of the culture grown on the same H₂/CO₂ concentrations as in Mix 1 and Mix 2 measured? This would seem to be necessary to decouple the effects of substrate concentration versus inhibitors.

Lines 88-91; Line 99: these values should all be reported in molarity, not g or ml per L.

Lines 124-126: It is not clear to me what this last sentence means in the context of the experiments, particularly Line 121 about H₂:CO₂ in a 4:1 ratio.

Line 180: Were OD measured in closed cuvetts? Anoxically? How were the cultures decompressed?

Line 193: What instrumentation was used to measure the gas pressure?

Line 219: conditions is misspelled

Line 251: It is not entirely clear to me from this description, was the pressure applied from the liquid medium or from the gas added (hydrostatic or hyperbaric)? This can have important consequences, especially with regards to decompression as shown in the work by Clark et al in the early 1990's.

Line 280: these OD measurements seem to be erroneous?

Reviewer #2 (Remarks to the Author):

Review of Taubner et al., "Biological methane production under Enceladus-like conditions"
Manuscript 138761_0_art_file2831601_yt0pw3

I applaud the efforts of the authors in studying the possibility of life in Enceladus ocean. However, there are several major concerns that must be addressed before this paper can be considered for publication.

The most important question to address in this review is whether the considered conditions are Enceladus-like, as implied by the title of this paper. The experimental conditions may not be similar to those inside Enceladus, but could be optimized for the growth of methanogens. There are two different issues that must be confronted to address this: 1) Is the data from Cassini INMS given in Extended Data Table 1 appropriately chosen, and 2) Has the plume data been properly extrapolated to the conditions of the internal ocean?

Table 1 lists the INMS data sets used in the study? The two most recent and up to date references do not even appear in the table. The most relevant data sets are found in reference 4 and reference 9. All other referenced data has been superseded by these publications. Use of any other data will likely introduce errors. With regard to the extrapolation of values from the plume to the ocean we find in Table 1, partial pressures of H₂ and CO₂. These values are significantly larger than the fugacities (~partial pressures) given in Supplementary Table S11 in Waite et al. (2017). It would be helpful to discuss this difference. Are the values in the submitted paper more appropriate for certain assumptions relating the plume composition to the ocean? How would the biological results be affected if the values from Waite et al. (2017) are the more appropriate values?

Another major objection is that the authors have not adequately referenced the results found in Waite et al. (2017). For example, the statement found in lines 30-33, "In addition, our kinetic..." are already established from measurements given in the Waite et al (2017) supplemental information.

Several other items call into question the assertion of Enceladus-like conditions in the experiments:

- 1.) The estimations of the internal structure of Enceladus could be strengthened considerably using constraints from recent papers analyzing geophysical data, such as Jess et al. (2014), McKinnon et al. (2015), and Thomas et al. (2016).
- 2.) Perhaps the most relevant temperature for Enceladus is 0 °C, which does not appear to be represented by the present experiments.
- 3.) Extended Figure 5 shows pH values from the experiments. In general, these values are much lower than estimates for Enceladus' ocean (see Table 1 in Glein et al., 2015), which may have a

pH between ~9 and ~11 (suggested by Waite et al., 2017). Preferably, an additional experiment should be performed at a more representative pH. I suggest that CO₂ gas not be added in this experiment, but add ~0.1 molar NaHCO₃ (Postberg et al., 2009) as the source of dissolved inorganic carbon.

Furthermore, I feel that the calculations of H₂ production by serpentinization are tacked on at the end of the text. If they are valuable to this paper, then their inclusion needs better justification. The concentrations of several species in the Enceladus ocean are assumed to be seawater-like. The consequences of this assumption on the computed H₂ production should be assessed.

Most notably it is unclear how H₂ production rates can be predicted from the rates of primary mineral dissolution. Ferrous iron needs to be oxidized by H₂O to produce H₂, not simply released to the solution. If dissolution rather than oxidation is rate-limiting for H₂ production, then this should be clarified and supported in the paper.

I suspect that the computed rate of H₂ production is very sensitive to the surface areas of the reacting minerals. These are unknown for the case of Enceladus. Therefore, specific values cannot be adopted. I suggest that the authors consider if ranges can be constrained based on data from meteorites, comets, and analogies to Earth. It would be useful to explain why particular values are selected.

In Table 2, CH₄ production rates are predicted. Are these biotic or abiotic? If the latter, I am deeply skeptical that methane can equilibrate with dissolved inorganic carbon at the considered temperature of 50 °C. The geochemical community considers abiotic production of methane to be kinetically inhibited, except under special circumstances (see the recent paper by McCollom, 2016, PNAS).

Reviewer #3 (Remarks to the Author):

The topic of this manuscript is relevant and exciting. Here are a few suggestions that I believe will make the manuscript better.

On lines 79, 110 and 199, there is discussion about Mix 1 and Mix 2. It would help if they were defined here, early in the manuscript, rather than having the reader have to search the tables. Lines 163 - 166 allude to measuring pressure in bottles. There is no detail about how this is done or what equipment is used. What is the source of the bottles? What is the source of the stoppers? In line 286, it mentions black stoppers. What source? Some stoppers, especially black stoppers, are notorious for leaking. Are the stoppers punctured to measure pressure? If so, how often? If often, how do they prevent leakage through the puncture holes?

In lines 175 - 178, it says pressure is measured in flasks. What type of flasks? Source of flasks? How was pressure measured?

In the Extended Data Table 3, there is no final volume mentioned for the solution.

In Figure 1 (a) and Extended Data Figure 1 (b, c), there are dips in the graphs, but no explanation.

Referee #1

The manuscript by Taubner, et al., “Biological methane production under Enceladus-like conditions” describes the results of a series of experiments aimed to measure the growth of autotrophic, hydrogenotrophic methanogens under conditions analogous to those predicted for the icy moon. The authors explore growth rates of 3 methanogenic species in the presence of different gas phases, chemistries, and even elevated hydrostatic pressures- that are expected to occur at the water/rock interface on Enceladus. The work considers the consequences of these variables in their ability to both support or inhibit the growth of the methanogenic species. The topic of habitability of Icy (or “Ocean”) Worlds of the outer solar system is extremely timely, building upon recent observational data and models of Enceladus’ interior and the contents of plumes emitted through cryovolcanism into space. I applaud the authors’ creative attempts to cultivate relevant model microorganisms under non-standard conditions such as may be expected on Enceladus. This is not often done, especially with challenging to grow strains and experimental rigor- both of which were tackled in the current study.

From my reading of this manuscript, the major claims of this paper are that 1) methanogenic organisms can function and produce methane under gas compositions and pressures expected to occur on Enceladus and 2) that potentially inhibitory compounds such as formaldehyde, ammonia, carbon monoxide, etc. do not severely limit growth at the concentrations tested. The data presented in this work appears to be thorough and replicable, and provides insight, at the least, on the physiology of methanogenic Archaea.

Thank you very much for the comments and for the summary of our work. We indeed performed our experiments thoroughly to be able to broadly understand methane production and growth kinetics of methanogens in different set-ups and at different pressure levels.

From my perspective, the work lacked several key elements that would make it important and interesting to a broad audience, beyond the archaeal physiology community. First, it is becoming more and more well-recognized that methane originating from serpentinizing environments on Earth is abiogenic. See recent works by David Wang (Science, 2015) and Giuseppe Etiope (e.g. Applied Geochemistry, 2017). While this does not exclude the contributions of biological methanogenesis to systems on either the modern-day Earth or Enceladus- the complete absence of a mention of biogenic versus abiogenic reaction mechanisms in the current manuscript is noticeable. Why might biogenic processes be more favorable or feasible on Enceladus than abiotic processes? How do known rates and pathways of biotic and processes compare in the physical/chemical environment visualized.

Thank you very much for your comments! We considered your comments carefully and added a results section on serpentinization to the manuscript’s text. In this section we specifically show results of our modelling. Furthermore, we discuss the limits of these simulations as well as the context of abiotic and biological methane production.

Second, I think that the authors missed a major opportunity to discuss how the growth of methanogens under Enceladus-like conditions could contribute to detectable biosignatures in the plumes.

Thank you very much for the great comment. We introduced a text on detection of biosignatures in the discussion section of the manuscript.

How many cells per liter could be supported (similar to the work by Chyba and Gaidos in the 1990's for Europa)? How do the activities of methanogens impact the composition of aqueous and gaseous metabolites?

Thank you very much for the questions. The calculation concerning how much C-fixed into biomass per H-fixed into biomass was based on the elemental mass balance data and molecular weight of methanogenic biomass. We also calculated a rate of methanogenic biomass production based on the serpentinization-based H₂ production rate. However, the assumptions used for the calculation of the biomass production rate relies on findings that were obtained under laboratory conditions and under optimal growth conditions and are likely to differ significantly from environmental conditions putative methanogens have to face on Enceladus. Hence, the calculation of the biomass production rate and the corresponding assumptions could be easily falsified and misinterpreted. However, we decided to present these rates because also calculations of biomass production *via* Gibbs energy dissipation resulted in similar numbers. Finally, we would like to add that autotrophic, hydrogenotrophic methanogens impact the composition of gaseous compounds while performing biological methane production.

Most importantly, in my opinion there should absolutely be a discussion of stable isotope signatures that have become a key point of focus for terrestrial systems with overlapping biogenic and abiogenic signatures. How might biological activities enhance or suppress these signatures. I am thinking specifically of the work by Ken Takai, et al (2008) studying hyperthermophilic methanogens at extremely high temperatures and pressures, showing a decrease in stable isotope fractionation. However, recent works by Wang, Sherwood Lollar, Etiope, and many others are equally relevant to the discussion.

Thank you very much for your comments. A full and comprehensive section on the detection of methanogenic biosignatures was added to the manuscript. In addition we considered the recently published selection of papers on Enceladus that appeared in a special issue in the journal *Astrobiology*.

In summary, the current manuscript is interesting and imaginative, and the data appears to be of high quality, thorough, and replicable. However, it also feels as if it only half-way met its potential, without a mention of how our interpretation of methane sources on Earth has been confounding, let alone in the outer solar system.

Thank you very much for the comments. We presented a thorough interpretation of our findings in respect to the literature on serpentinization and abiotic as well as biological methanogenesis on Earth. We hope that the potential readers of our manuscript can now integrate our results to the confounding reports on abiotic and biological methane production on planet Earth into the experimental setting and the serpentinization modelling applied in this study.

Specific edits and corrections are mentioned below:

Line 80: Were the growth kinetics of the culture grown on the same H₂/CO₂ concentrations as in Mix 1 and Mix 2 measured? This would seem to be necessary to decouple the effects of substrate concentration versus inhibitors.

The concentrations of H₂/CO₂ in the liquid phase are shown in Table 3. In the series of experiments we decided to use H₂/CO₂ as positive control because the intention was not to investigate the relative or absolute effect of the inhibitors on the physiology of the organism but to examine whether methanogen(s) could survive, grow, and produce methane under the tested conditions. Generally H₂/CO₂ is used as a standard gas for cultivation of autotrophic, hydrogenotrophic methanogens in pure culture. The results of the experiments should in our opinion be relatable to standard laboratory conditions.

Lines 88-91; Line 99: these values should all be reported in molarity, not g or ml per L.

In an earlier version of our manuscript we already presented most of the units in molarities. Considering your comment we present now the other units also in molarities. However, for formaldehyde we cannot present the concentration of this compound in mol L⁻¹ because CH₂O at room temperature does not exist in liquid state.

Lines 124-126: It is not clear to me what this last sentence means in the context of the experiments, particularly Line 121 about H₂:CO₂ in a 4:1 ratio.

We detected H₂/CO₂ to CH₄ conversion in all high pressure experiments, both with H₂/CO₂ gas phase and H₂/CO₂/N₂ gas phase, except for the H₂/CO₂ experiment at 90 bar where no gas conversion could be observed.

Line 180: Were OD measured in closed cuvetts? Anoxically? How were the cultures decompressed?

OD was measured oxically in open cuvettes (Semi-micro cuvette, 1.6 ml made of Polystyrene, Sarstedt, Germany). The samples were taken from the serum bottles without any specific decompression step, because we haven't detected any decompression effects on the cells at low pressure (below 3 bar) in any of our experiments. No OD measurements were performed during high pressure experiments because cell disruption occurs upon depressurization.

Line 193: What instrumentation was used to measure the gas pressure?

As mentioned in in Taubner, R.-S. & Rittmann, S. K.-M. R. “Method for indirect quantification of CH₄ production via H₂O production using hydrogenotrophic methanogens”. *Front. Microbiol.* 7, (2016)), we used a digital manometer (LEO1-Ei, -1...3bar rel, Keller, Germany) to measure the serum bottle headspace pressure during experiments up to 3 bar.

Line 219: conditions is misspelled

The error has been corrected.

Line 251: It is not entirely clear to me from this description, was the pressure applied from the liquid medium or from the gas added (hydrostatic or hyperbaric)? This can have important consequences, especially with regards to decompression as shown in the work by Clark et al in the early 1990's.

Thank you very much for the comment. The pressure in our experiments was hyperbaric. The study by Clark *et al.* on the growth activity by applying either hyperbaric or hydrostatic pressure on a stem originating from a deep sea hydrothermal vent, showed higher specific growth rates for the hydrostatic culture compared to the experiments in the hyperbaric vessel. However, in the experiments by Clark *et al.* no gas phase was present whereas the consumed or produced gases in our study could be held relatively constant in a hyperbaric closed batch system.

Line 280: these OD measurements seem to be erroneous?

The OD measurements were reported correctly. To receive such a high OD, we used approximately 1L of fresh culture (OD ~0.7), centrifuged the cultivation suspension anaerobically, and re-suspended the cells in 20 mL of freshly reduced appropriate growth medium. This information has now been added to the method section.

Referee #2

I applaud the efforts of the authors in studying the possibility of life in Enceladus ocean. However, there are several major concerns that must be addressed before this paper can be considered for publication. The most important question to address in this review is whether the considered conditions are Enceladus-like, as implied by the title of this paper.

Thank you for the comments. We changed the title to “Biological methane production under putative Enceladus-like conditions”. We considered the conditions in our experiments as close as one can get to putative Enceladus-like in the laboratory. We relied on results published on the physicochemical conditions of Enceladus for designing our experiments. Hence, the conditions used in all our experiments were selected according to the state of the knowledge and intended to be as close as possible to putative Enceladus-like conditions.

The experimental conditions may not be similar to those inside Enceladus, but could be optimized for the growth of methanogens. There are two different issues that must be confronted to address this: 1) Is the data from Cassini INMS given in Extended Data Table 1 appropriately chosen, and 2) Has the plume data been properly extrapolated to the conditions of the internal ocean? Table 1 lists the INMS data sets used in the study? The two most recent and up to date references do not even appear in the table. The most relevant data sets are found in reference 4 and reference 9. All other referenced data has been superseded by these publications. Use of any other data will likely introduce errors. With regard to the extrapolation of values from the plume to the ocean we find in Table 1, partial pressures of H₂ and CO₂. These values are significantly larger than the fugacities (~partial pressures) given in Supplementary Table S11 in Waite et al. (2017). It would be helpful to discuss this difference.

Thank you very much for these critical comments! We agree with your comment that our partial pressures are significantly larger than those reported by Waite *et al.* 2017. These high values are caused by the usage of previous Cassini data (e.g. presented in Waite et al., 2009). We would suggest interpreting these values as upper limits for biological methane production under putative Enceladus-like condition. Furthermore, we included the new INMS results published by Waite *et al.* 2017 in Table 1. We show that *M. okinawensis* was able to withstand the mentioned physicochemical conditions at even these higher concentrations as the reported by Waite *et al.* 2017. Lower concentrations of inhibitors did not measurably affect the physiology of the tested methanogens under laboratory conditions. Eventually we would like to add, that to be able to design experiments suited to address at which detailed physicochemical conditions methanogens would need to propagate and produce CH₄ more information on the astrogeological properties of Enceladus is needed. Until then, the experimental approach used here was tailored to narrow down and evidence that biological CH₄ production under the putative inferred Enceladus-like conditions is in principle possible.

Are the values in the submitted paper more appropriate for certain assumptions relating the plume composition to the ocean? How would the biological results be affected if the values from Waite et al. (2017) are the more appropriate values?

Thank you for the comment! As mentioned before, the data should be seen as upper concentration limits. We added the following sentence to the manuscript: “The mixing ratios of these putative inhibitors were based on INMS data^{4,5,25,26} but higher than those calculated by using the most recent Cassini data^{2,6} (Table 1). This demonstrates that even at higher inhibitor concentrations growth and biological CH₄ production of *M. okinawensis* is possible..”

Another major objection is that the authors have not adequately referenced the results found in Waite et al. (2017). For example, the statement found in lines 30-33, “In addition, our kinetic....” are already established from measurements given in the Waite et al (2017) supplemental information.

Thank you for the comment. We added a full results section on serpentinization and also added the corresponding material and methods as well as discussion section to the manuscript. We specifically indicated the proposed reference as introductory sentence in this specific section. We hope that all the relevant and recent literature, which supports our proposed serpentinization model, has now been included and was adequately used whenever needed throughout the manuscript’s text.

Several other items call into question the assertion of Enceladus-like conditions in the experiments:

1.) The estimations of the internal structure of Enceladus could be strengthened considerably using constraints from recent papers analyzing geophysical data, such as Iess et al. (2014), McKinnon et al. (2015), and Thomas et al. (2016).

Thank you very much for your comment! Thomas et al. (2016) estimated a global subsurface ocean with a south polar depression based on the moment of inertia reported by Iess et al. (2014). They predict an ocean thickness of 26-31 km and an icy crust of 26-21 km with just 13 km under the SPT. The considered densities of the three layers are 2300, 1000, and 850 kg m⁻³ in the model by Thomas et al. (2016) are therefore corresponding to our estimations. In our previous estimations, the ice shell is set to 30 to 40 km, which might be too thick regarding the newest findings. However, according to our calculations, the hydrostatic pressure varies just by some bars using the newest data. We modified the manuscript to address the reviewer’s comments as follows: “Calculations of the hydrostatic pressure based on Enceladus mass of 1.0794·10²⁰ kg^[42] and its mean radius of 252.1 km^[42] assuming a core radius of 190-200 km, an ocean depth of 60-10 km and a corresponding ice shell thickness of 2.1-52.1 km results in a pressure of approx. 44.3- 25.2 bar at the water-core-boundary. For the high pressure experiments in this study including all inhibitors, three pressure values were

chosen that lie in the range given by Hsu et al. (10-80 bar)⁸ and are related to our calculations, *i.e.* 10, 25, and 50 bar.”

2.) *Perhaps the most relevant temperature for Enceladus is 0 °C, which does not appear to be represented by the present experiments.*

Thank you for the comment! The assumptions regarding temperature are based on Hsu, H.-W. et al. “Ongoing hydrothermal activities within Enceladus”, *Nature* (2015) 519:207–210. In that article, the authors report on the likelihood of ongoing hydrothermal activities with an accompanied raise in temperature to more than 90°C. We assume that if there is biotic activity on Enceladus, then it will be situated in the surrounding area of these hydrothermal vents (Barge and White, 2017). Further, working with methanogens at temperature near 0°C is possible (e.g. with *Methanosarcina soligelidi* or *Methanogenium frigidum*) but extremely time consuming because not only low specific growth rates are characteristic for these organisms, but in addition we apply gas-limited growth conditions. Due to this reasoning and due to the fact that hydrothermal activity is suggested to be ongoing in Enceladus we decided to perform the experiments with thermophilic and hyperthermophilic organisms. We added some of these comments and reply to the new discussion part of the manuscript.

3.) *Extended Figure 5 shows pH values from the experiments. In general, these values are much lower than estimates for Enceladus’ ocean (see Table 1 in Glein et al., 2015), which may have a pH between ~9 and ~11 (suggested by Waite et al., 2017). Preferably, an additional experiment should be performed at a more representative pH. I suggest that CO₂ gas not be added in this experiment, but add ~0.1 molar NaHCO₃ (Postberg et al., 2009) as the source of dissolved inorganic carbon.*

Thank you for the highly valuable comments! Do to reasons of experimental reproducibility we always used CO₂ as the sole source of carbon for growth of methanogens. To overcome the pH drift due to high CO₂ partial pressure and associated dissolved CO₂ in the medium, using NaHCO₃ as the source of dissolved inorganic carbon would be a possibility. Postberg *et al.* (2009) suggested a concentration of 0.02-0.1 mol kg⁻¹ NaCO₃ and 0.05-0.2 mol kg⁻¹ NaCl in the medium to reach a pH level between 8.5 and 9. Calculations on the concentrations of dissolved CO₂ in the high pressure experiments were performed by using the mole fraction of dissolved CO₂ in water depending on the CO₂ partial pressure ($p(\text{CO}_2)$) (CRC handbook of chemistry and physics, 2004). The mole fractions for the CO₂ partial pressures of 0.7, 1.2 and 3.1 bar (used in the experiments) at 65 °C were generated by extrapolation of given values in the region of 0-1 bar CO₂ partial pressure. The resulting NaHCO₃ concentrations $c(\text{NaHCO}_3)$ corresponding to the given $p(\text{CO}_2)$ in the experiments would lead to a shift of pH to 9.2-9.5 since NaHCO₃ is dissociating into CO₂ and OH⁻ ions (Storhas *et al.*, 2013). These pH levels would fit to the estimated values for the subsurface ocean on Enceladus (Zolotov *et al.* (2009)), as indicated in the manuscript. Values for the NaHCO₃ concentrations and interfering pH for the corresponding CO₂ partial pressures:

$p(\text{CO}_2)$ / bar	mole fraction x 1000	$c(\text{NaHCO}_3)$ / mM	$c(\text{NaHCO}_3)$ / g L ⁻¹	pH
0.7	0.196	10.9	0.9	9.2
1.2	0.333	18.5	1.6	9.3
3.1	0.846	47.0	3.9	9.5

However, our own computations indicated a pH range in the ocean of Enceladus of 7.3-11.3. We are aware of the pH discrepancy between our experiments and the Enceladian values estimated by our or models and by Waite *et al.* 2017. Experiments without any added CO_2 were already performed in the initial version of the manuscript submitted before (see Figure 1 (g,h,i)), but for *M. okinawensis* little growth was detected in exactly these experiments. *M. okinawensis* was able to produce CH_4 even at a pH of 3.5 in experiments at 50 bar with H_2/CO_2 (4:1), *i.e.* $p_{\text{CO}_2} = 10$ bar (Table 4 and Supplementary Table 4), which is one magnitude below the earlier reported lower pH limit of 4.5 of this methanogen. This showed the metabolic versatility and the yet unexplored extremophilic capabilities of this organism. Furthermore, during the 2 bar liquid and gaseous inhibitor experiments using *M. okinawensis* the pH was in the range of 7-8, which is in the lower pH range of our model. *M. marburgensis* showed good growth on carbonates only in the presence of H_2 if one considers the newest concentrations of inhibitors and the pressure levels on Enceladus. Hence, *M. marburgensis* could be an organism of interest to perform carbonate based high pH experiments and high pressure CO_2/H_2 to CH_4 conversion. Apart from *M. marburgensis* there are several known methanogenic strains that can grow at high pH values (9-10.2), from which we might select some strains for future experiments. We added some of these thoughts to the newly introduced discussion section and the methods part of the manuscript.

The concentrations of several species in the Enceladus ocean are assumed to be seawater-like. The consequences of this assumption on the computed H_2 production should be assessed. Thank you for the comment! The composition of seawater has no bearing on the modelled H_2 production rate. This is because the water-to-rock ratios are exceedingly low, so that the fluid composition will be entirely rock-buffered.

Most notably it is unclear how H_2 production rates can be predicted from the rates of primary mineral dissolution. Ferrous iron needs to be oxidized by H_2O to produce H_2 , not simply released to the solution. If dissolution rather than oxidation is rate-limiting for H_2 production, then this should be clarified and supported in the paper.

Thank you for the interesting comment! Going into which reaction is the rate-limiting step in H_2 production during serpentinization would be opening a can of worms. It would go beyond the scope of this paper. We reference a paper that discusses this rationale for this assumption in kinetic modelling in greater detail. We have inserted the following sentence to our

manuscript to address your comment: “Of the many reactions involved in serpentinization of peridotite, dissolution of the Fe(II)-bearing primary phases is a critical one²⁹, and the only one for which kinetic data is available.”

I suspect that the computed rate of H₂ production is very sensitive to the surface areas of the reacting minerals. These are unknown for the case of Enceladus. Therefore, specific values cannot be adopted. I suggest that the authors consider if ranges can be constrained based on data from meteorites, comets, and analogies to Earth. It would be useful to explain why particular values are selected.

Thank you very much for the comment! The following line of argumentation has been inserted to the manuscript in response to your suggestion. These specific surface areas have been suggested to be typical for fine-grained terrestrial rocks. We adopted these numbers in our computations, as we have no constraints on what specific surface areas in Enceladus may be. If the core of Enceladus was similar to carbonaceous chondrite, then the average mineral grain size is smaller and hence the specific surface areas greater than we assumed. We choose to use fairly small specific surface areas to provide conservative estimates for H₂ production rates.

In Table 2, CH₄ production rates are predicted. Are these biotic or abiotic? If the latter, I am deeply sceptical that methane can equilibrate with dissolved inorganic carbon at the considered temperature of 50 °C. The geochemical community considers abiotic production of methane to be kinetically inhibited, except under special circumstances (see the recent paper by McCollom, 2016, PNAS).

Thank you for your comment! The sentence “If H₂ levels in the model system are high enough to provide driving force for CO₂ reduction, then the model predicts that CH₄ production is possible”, which could be found in an earlier version of this manuscript in lines 164-170. The text was adapted and replaced by the following text: “In the model, CO₂ reduction to CH₄ is predicted to take place once enough H₂ in the system was produced to generate thermodynamic drive for the reaction. While abiotic CH₄ production is kinetically more sluggish than olivine dissolution³⁰, biological CH₄ production is fast and may be controlled by the rate at which H₂ is supplied. The abiotic CH₄ production rates listed in Table 5 are hence also modelled such that olivine dissolution is the rate limiting step.”

Referee #3

The topic of this manuscript is relevant and exciting. Here are a few suggestions that I believe will make the manuscript better.

Thank you very much for reviewing our manuscript!

On lines 79, 110 and 199, there is discussion about Mix 1 and Mix 2. It would help if they were defined here, early in the manuscript, rather than having the reader have to search the tables.

Thank you for the comment. We added the composition and a hint to Table 2: “As expected, the final optical densities did not reach those of the experiments with H₂/CO₂, likely because in Mix 1 and Mix 2 lower absolute amounts of convertible gaseous substrate (H₂/CO₂) were available for conversion compared to the growth under pure H₂/CO₂. Consequently, growth kinetics showed a different, gas-limited linear inclination in the closed batch setup when using Mix 1 and Mix 2^{22,24}”.

Lines 163 - 166 allude to measuring pressure in bottles. There is no detail about how this is done or what equipment is used. What is the source of the bottles? What is the source of the stoppers?

For this procedure, we used the methods, consumables, and instruments described in Ref 24 (Taubner, R.-S. & Rittmann, S. K.-M. R. Method for indirect quantification of CH₄ production via H₂O production using hydrogenotrophic methanogens. *Front. Microbiol.* 7, (2016)). We used a digital manometer (LEO1-Ei, -1...3bar rel, Keller, Germany) to measure the serum bottle headspace pressure. Also mentioned in that reference, we used 120 mL serum bottles (La-Pha-Pack, Langerwehe, Germany), sealed with blue rubber stoppers (pretreated by boiling ten times for 30 min in fresh ddH₂O; 20mm, butyl rubber, CLS-3409-14, Chemglass Life Sciences) and crimp caps (20mm aluminum, Ochs Laborbedarf, Bovenden, Germany).

In line 286, it mentions black stoppers. What source? Some stoppers, especially black stoppers, are notorious for leaking. Are the stoppers punctured to measure pressure? If so, how often? If often, how do they prevent leakage through the puncture holes? In lines 175 - 178, it says pressure is measured in flasks. What type of flasks? Source of flasks? How was pressure measured?

Thank you for the comments. The black stoppers used for the gas samples during the high pressure experiments were black septa (Butyl/PTFE, La Pha Pack, Langerwehe, Germany). We added this information to the manuscripts text. The septa were always new and therefore maximally 2 times punctured when GC analysis took place. Before GC measurements, the septa were always bulging out which is a sign for overpressure in the flasks and therefore for

the leak proofness of the septa. For the experiments at low pressure we used the blue rubber stoppers mentioned before.

In the Extended Data Table 3, there is no final volume mentioned for the solution.

Thank you for the comment! We added the amount of the final volume (1L) in the caption.

In Figure 1 (a) and Extended Data Figure 1 (b, c), there are dips in the graphs, but no explanation.

Thank you for the comment! The dips in the graph were caused by substrate limitation due to depletion of serum bottle headspace of H₂/CO₂ at high optical cell densities. We added this information to the figure captions.

REVIEWERS' COMMENTS:

Reviewer #1 (Remarks to the Author):

The revised manuscript by Taubner, et al. represents a greatly improved version of the earlier submission. Many of the concerns I raised in an earlier review of this manuscript were addressed, including adding technical details and a discussion of the geochemical production of methane via serpentinization. I especially appreciated the discussion of biosignatures that the authors added to the end of the manuscript, as this serves to link their models of extraterrestrial habitability to potentially detectable compounds in near term space missions.

Specific comments are below:

Line 25: Aren't Pascals the appropriate SI unit here?

Line 50: temperatures and pressure should be plural

Line 51: Is aquifer the correct term here? Are you talking about a subsurface "ocean" or hydrated pore spaces in the solid interior?

Lines 122, 129, 131, 133 and throughout: should be SI units (Pascals) again.

Lines 175: I agree with this, but also that potential biological processes would compete with abiotic processes, and that the resultant methane is likely to carry a mixed signal.

Line 186: I am not entirely sure what you are getting at here. There is clearly a discrepancy between modeled pH on Enceladus and the pH at which the experiments were conducted. I presume that what you are trying to say is that future experiments could use bicarbonate (HCO_3^-) as a carbon source for methanogenesis and thereby circumvent the problem of acidification in the medium.

Discussion: I really like the text and considerations you have added with regards to methanogenic biosignatures and methane isotopes. This puts the experimental work into much better context.

Line 295: It is still not entirely described in the methods how pressure measurements were done.

Line 371: not sure what you are getting at here. Why is N_2 fixation assumed?

Line 415: this makes it sound like some sort of layered septum- Is that the case? Or were these your typical 20 mm butyl rubber septa (as used in Balch tubes and similar incubations).

Line 669: here you talk about a hydrostatic pressure

Figure 2: caption is very obtuse. I presume that the color scheme in the 3-dimensional spot correspond to the growth curves presented in the plot. However, it would be extremely worthwhile if this was articulated in the caption itself. Otherwise, it is hard to make sense of the figure.

Reviewer #2 (Remarks to the Author):

Reviewer #2 (Remarks to the Author):

The paper has definitely been improved by the responsiveness of the authors.

Major comments

One remaining issue is the consistency of the H₂ production model results. An estimated production rate of (3-40)x10⁴ mol H₂/y is given. However, this is much lower than the range of (1-5)x10⁹ mol H₂/y derived from Cassini data (Waite et al. 2017). These values should be reconciled. Maybe consider water flow rates from Choblet et al. (2017)?

Below Table 3, it is stated that the pressure in the INMS instrument is 1 bar. This is incorrect. Not clear if this case is important, as the gases coming into INMS are those that have been degassed and transported from the ocean, rather than gases currently in equilibrium with the ocean.

The biomass production rates given in the paper seem opaque. I think it would help to give some context for those who are not biologists. For example, how do these values compare to hydrothermal ecosystems on Earth?

I am not understanding the below pH estimates given in the rebuttal

$p(\text{CO}_2)$ / bar	mole fraction x 1000	$c(\text{NaHCO}_3)$ / mM	$c(\text{NaHCO}_3)$ / g L ⁻¹	pH
0.7	0.196	10.9	0.9	9.2
1.2	0.333	18.5	1.6	9.3
3.1	0.846	47.0	3.9	9.5

Consider the following equilibrium

For representative values of 1 bar CO₂ and 0.01 molal HCO₃⁻, I obtain an approximate (ideal) pH of 5.7, as might be expected for a high CO₂ fugacity system. Not sure where pH near 9 comes from.

It would be useful to clarify this discrepancy to make sure that the paper treats gas-liquid equilibrium correctly.

Minor comments

An issue that is emerging in the discussion of biological methane production on Enceladus is the relatively large abundance of H₂ in the plume. Some consider this observation to be a potential anti-biosignature, because it is argued that if methanogens were present, they would deplete the H₂ to a much lower abundance. Given that this paper is advocating for biology, it would be useful to confront this issue with some discussion. Perhaps slow growth at low temperatures, a nickel limitation, or habitats restricted to near hydrothermal vents?

The ocean floor pressure computed in this work seems anomalously low. The pressure can be approximated using an ocean density of 1000 kg m⁻³, a gravitational acceleration of 0.113 m s⁻², and a hydrosphere thickness of 60 km. This leads to an ocean floor pressure of about 70 bar.

It could be of interest to briefly compare the H₂/CH₄ ratio predicted from the modeling to the ratio observed in the plume.

Footnote for Table 5 – should read dissolved inorganic carbon

There seems to be a misunderstanding that the upper limit temperature of hydrothermal systems on Enceladus is 90-100 C. This is actually a lower limit from Hsu et al. (2015). As far as I am aware, there is no observationally based upper limit. Of course, even if hydrothermal vent fluids are much hotter, a range of lower temperatures would occur in mixing zones around the vents.

Reviewer #3 (Remarks to the Author):

Thank you for addressing my concerns. I believe the manuscript is acceptable for publication.

Response to editor and reviewer comments

Referee #1

The revised manuscript by Taubner, et al. represents a greatly improved version of the earlier submission. Many of the concerns I raised in an earlier review of this manuscript were addressed, including adding technical details and a discussion of the geochemical production of methane via serpentinization. I especially appreciated the discussion of biosignatures that the authors added to the end of the manuscript, as this serves to link their models of extraterrestrial habitability to potentially detectable compounds in near term space missions.

Specific comments are below:

Line 25: Aren't Pascals the appropriate SI unit here?

Thank you very much for the comment. Even though Pascal is the SI unit, we prefer to use the entity bar. Bar is a very common unit, may be directly converted into Pa and the readability is more useful due to missing magnitudes.

Line 50: temperatures and pressure should be plural

We changed the respective wording into plural.

Line 51: Is aquifer the correct term here? Are you talking about a subsurface “ocean” or hydrated pore spaces in the solid interior?

We changed the word “aquifer” to “subsurface ocean” throughout the manuscript.

Lines 122, 129, 131, 133 and throughout: should be SI units (Pascals) again.

See comment before.

Lines 175: I agree with this, but also that potential biological processes would compete with abiotic processes, and that the resultant methane is likely to carry a mixed signal.

Thank you very much for the comment. We added this reasoning to the text.

Line 186: I am not entirely sure what you are getting at here. There is clearly a discrepancy between modeled pH on Enceladus and the pH at which the experiments were conducted. I presume that what you are trying to say is that future experiments could use bicarbonate (HCO_3^-) as a carbon source for methanogenesis and thereby circumvent the problem of acidification in the medium.

Thank you very much for the comment. We deleted this text passage from the manuscript.

Discussion: I really like the text and considerations you have added with regards to methanogenic biosignatures and methane isotopes. This puts the experimental work into much better context. Line 295: It is still not entirely described in the methods how pressure measurements were done.

Thank you very much for the comment. As described in ref 22 (Taubner & Rittmann, 2016), to measure the serum bottle head space pressure we used a digital manometer (LEO1-Ei, -1...3 bar rel, Keller, Germany). The proceeding can be seen in Fig. 3(A) of this article. We added this information to the text.

Line 371: not sure what you are getting at here. Why is N₂ fixation assumed?

Thank you very much for the comment. The effect of N₂ fixation was investigated by elementary analysis of biomass obtained from fermentations at 20 bar with the gas substrates H₂:CO₂=4:1 or H₂:CO₂:N₂=4:1:5. The biomass of the fermentation with additional N₂ in the gas phase showed higher nitrogen content in contrary to the experiment performed without N₂ in the gas phase (See Table 1). Therefore, a dependency between the N-content in the biomass and the presence of N₂ in the substrate gas was assumed. However, the N-content in the biomass could not be performed in duplicates due to an insufficient amount of the sample.

Table 1: N-content in the biomass depending on whether N₂ was present in the substrate gas or not.

partial pressures of gases			N-content in biomass / m%
N ₂ / bar	H ₂ / bar	CO ₂ / bar	
0	16	4	8.115
10	8	2	9.259

Line 415: this makes it sound like some sort of layered septum- Is that the case? Or were these your typical 20 mm butyl rubber septa (as used in Balch tubes and similar incubations).

Thank you very much for the comment. The butyl septa we used for the storage of the gas samples are simple butyl septum as used for GPC-vials. We added the thickness of the butyl septa to the manuscript.

Line 669: here you talk about a hydrostatic pressure

We refer to hydrostatic pressure.

Figure 2: caption is very obtuse. I presume that the color scheme in the 3-dimensional spot correspond to the growth curves presented in the plot. However, it would be extremely worthwhile if this was articulated in the caption itself. Otherwise, it is hard to make sense of the figure.

Thank you very much for the comment. As already stated in the caption, the colours of the dots and the letters of the figure in the upper left corner correspond to the growth curves. They were grouped regarding their similar behaviour in the increase of the OD during the experiment.

Referee #2

The paper has definitely been improved by the responsiveness of the authors.

Major comments

One remaining issue is the consistency of the H₂ production model results. An estimated production rate of (3-40)×10⁴ mol H₂/y is given. However, this is much lower than the range of (1-5)×10⁹ mol H₂/y derived from Cassini data (Waite et al. 2017). These values should be reconciled. Maybe consider water flow rates from Choblet et al. (2017)?

In response to this helpful comment we added the following description to state the discrepancy offer an explanation: " Our computed steady state H₂ production rates are lower than the 1-5 × 10⁹ mol H₂ y⁻¹ estimated from Cassini data². This apparent discrepancy in flux rates can be reconciled if the Enceladus plume was a transient (i.e. non-steady state) phenomenon".

Below Table 3, it is stated that the pressure in the INMS instrument is 1 bar. This is incorrect. Not clear if this case is important, as the gases coming into INMS are those that have been degassed and transported from the ocean, rather than gases currently in equilibrium with the ocean.

We removed this half-sentence from Table 1.

The biomass production rates given in the paper seem opaque. I think it would help to give some context for those who are not biologists. For example, how do these values compare to hydrothermal ecosystems on Earth?

Thank you for the comment. AS we specified in the manuscript the rates were calculated in a transparent way by using published elementary composition of methanogenic biomass and on the other hand thermodynamic data for CH₄ formation of methanogens.

I am not understanding the below pH estimates given in the rebuttal

$p(\text{CO}_2)$ / bar	mole fraction x 1000	$c(\text{NaHC}$ $\text{O}_3) / \text{mM}$	$c(\text{NaHCO}_3) / \text{g L}^{-1}$	pH
0.7	0.196	10.9	0.9	9.2
1.2	0.333	18.5	1.6	9.3
3.1	0.846	47.0	3.9	9.5

Consider the following equilibrium

For representative values of 1 bar CO₂ and 0.01 molal HCO₃⁻, I obtain an approximate (ideal) pH of 5.7, as might be expected for a high CO₂ fugacity system. Not sure where pH near 9 comes from. It would be useful to clarify this discrepancy to make sure that the paper treats gas-liquid equilibrium correctly.

The pH values were not calculated for a buffered system with CO₂ and NaHCO₃. The pH was calculated for the NaHCO₃ solution (ideal) that should be used, to achieve the same concentrations of dissolved CO₂ in H₂O as in the experiments where CO₂ was supplied as inorganic carbon source.

Calculations on the concentrations of dissolved CO₂ in the high pressure experiments were performed by using the mole fraction of dissolved CO₂ in water depending on the CO₂ partial pressure ($p(\text{CO}_2)$), CRC handbook of chemistry and physics, A ready-reference book of chemical and physical data, Ed.: Lide, D. R., (2004), 85. ed., CRC Press, Boca Raton.). The mole fractions for the CO₂ partial pressures of 0.7, 1.2 and 3.1 bar that were used in the experiments at 65 °C were generated by extrapolation of given values in the region of 0-1 bar CO₂ partial pressure. The resulting NaHCO₃ concentrations $c(\text{NaHCO}_3)$ corresponding to the given $p(\text{CO}_2)$ in the experiments would lead to a shift of pH to 9.2-9.5 since NaHCO₃ is dissociating into CO₂ and OH ions (Table 1) (Storhas W., Behrendt U., Rubbeling H., Wiedemann P., Bioverfahrensentwicklung, (2013), 2., vollst. überarb. und aktualisierte Aufl., Wiley-VCH; Wiley, Weinheim, Hoboken, NJ.). These pH levels would fit to the estimated values for the subsurface ocean on Enceladus by Zolotov et al. in 2009.

Minor comments

An issue that is emerging in the discussion of biological methane production on Enceladus is the relatively large abundance of H₂ in the plume. Some consider this observation to be a potential anti-biosignature, because it is argued that if methanogens were present, they would deplete the H₂ to a much lower abundance. Given that this paper is advocating for biology, it would be useful to confront this issue with some discussion. Perhaps slow growth at low temperatures, a nickel limitation, or habitats restricted to near hydrothermal vents?

We favor the latter explanation. We expect that methanogenesis to be confined to what one could refer to as biogeochemical hotspots (i.e. hydrothermal vents). On Earth, one would find signatures of methanogens only in direct proximity to such hot spots, for example in the chimney walls of aragonite/brucite chimneys of the Lost City Hydrothermal Field, but not in surrounding rocks or sediments (Méhay et al., 2013, Geobiology, DOI:10.1111/gbi.12062). We actually make a statement to this end in the manuscript. However, the other explanations could also be considered. With respect to low temperatures biological methane production, the specific growth rates of psychrophilic methanogens and, hence the specific methane productivities are with exceptions known, much lower compared to thermophilic or hyperthermophilic methanogens.

The ocean floor pressure computed in this work seems anomalously low. The pressure can be approximated using an ocean density of 1000 kg m⁻³, a gravitational acceleration of 0.113 m s⁻², and a hydrosphere thickness of 60 km. This leads to an ocean floor pressure of about 70 bar.

Thank you very much for your comment. We recalculated the hydrostatic pressure with two other different methods and in one case we received similar numbers. We changed the respective text phrases accordingly and included the code in the Supplementary Material.

It could be of interest to briefly compare the H₂/CH₄ ratio predicted from the modeling to the ratio observed in the plume.

We added the following sentence: “The predicted H₂/CH₄ ratio of 2.5 (Fo₉₀:En:Diop = 8:1:1) to 4 (Fo₉₀) for the magnesian compositions of Enceladus’ core (Table 5) are consistent with the relative proportions of the two gases in the plume (0.4-1.4% H₂, 0.1-0.3% CH₄)²”.

Footnote for Table 5 – should read dissolved inorganic carbon.

Thank you very much for the comment. We changed it accordingly.

There seems to be a misunderstanding that the upper limit temperature of hydrothermal systems on Enceladus is 90-100 C. This is actually a lower limit from Hsu et al. (2015). As far as I am aware, there is no observationally based upper limit. Of course, even if hydrothermal vent fluids are much hotter, a range of lower temperatures would occur in mixing zones around the vents.

We changed it accordingly by adding the words “above” or “higher than” to the corresponding sentences in the manuscript.

Referee #3

Thank you for addressing my concerns. I believe the manuscript is acceptable for publication.

Thank you very much!